# Effects of *Marquandomyces marquandii* SGSF043 on the Germination Activity of Chinese Cabbage Seeds: Evidence from Phenotypic Indicators, Stress Resistance Indicators, Hormones and Functional Genes

**DOI:** 10.3390/plants14010058

**Published:** 2024-12-27

**Authors:** Xu Zheng, Yuxia Huang, Xinpeng Lin, Yuanlong Chen, Haiyan Fu, Chunguang Liu, Dong Chu, Fengshan Yang

**Affiliations:** 1Engineering Research Center of Agricultural Microbiology Technology, Ministry of Education & Heilongjiang Provincial Key Laboratory of Ecological Restoration and Resource Utilization for Cold Region & Key Laboratory of Molecular Biology, College of Heilongjiang Province & School of Life Sciences, Heilongjiang University, Harbin 150080, China; ztlovezx@163.com (X.Z.); huangyuxia2003@126.com (Y.H.); lz_xinpeng@163.com (X.L.); 2221630@s.hlju.edu.cn (Y.C.); 2005013@hlju.edu.cn (C.L.); 2College of Plant Health and Medicine, Qingdao Agricultural University, Qingdao 266109, China; chinachudong@qau.edu.cn

**Keywords:** *Metarhizium*, *Marquandomyces marquandii*, Chinese cabbage, gene expression

## Abstract

In this study, the effect of *Metarhizium* spp. *M. marquandii* on the seed germination of cabbage, a cruciferous crop, was investigated. The effects of this strain on the seed germination vigor, bud growth and physiological characteristics of Chinese cabbage were analyzed by a seed coating method. The results showed the following: (1) The coating agent *M. marquandii* SGSF043 could significantly improve the germination activity of Chinese cabbage seeds. (2) The strain concentration in the seed coating agent had different degrees of regulation on the antioxidase system of the buds, indicating that it could activate the antioxidant system and improve the antioxidant ability of the buds. (3) When the concentration of *M. marquandii* SGSF043 was 5.6 × 10^6^ CFU/mL (average per grain), the effect of *M. marquandii* SGSF043 on the leaf hormones Indole Acetic Acid (IAA), Gibberellic Acid (GA) and Abscisic Acid (ABA) of Chinese cabbage seedlings was significantly higher than that of other treatment groups, indicating that the strain could optimize the level of plant hormones. (4) *M. marquandii* SGSF043 could induce the expression of stress-resistance-related genes in different tissue parts of Chinese cabbage and improve the growth-promoting stress resistance of buds. This study showed that *M. marquandii* SGSF043 could not only improve the germination vitality of Chinese cabbage seeds but also enhance the immunity of young buds. The results provide a theoretical basis for the application potential of *Metarhizium marquandii* in agricultural production.

## 1. Introduction

*Brassica rapa* belongs to the Brassica genus of the cruciferous family. As an important leaf vegetable crop, *B. rapa* has attracted wide attention worldwide. However, biological and abiotic stresses have a serious impact on the quality and quantity loss of cabbage yield [1,2]. In Integrated Pest Management (IPM) and the development of organic agriculture system actions and strategies, emphasis is placed on the use of microorganisms and their metabolites to enhance plant immunity to biological/abiotic stresses and to promote crop growth [3,4,5,6]. Fang et al. [7] analyzed the rhizosphere, root surface and root microbial community composition of disease-resistant brassica rape varieties, indicating that microorganisms can maintain their biodiversity and microbial community function through plant root exudates. For example, *Flavobacteria* and *Sphingosphinomonas* maintain strong metabolic capacity and are in a favorable position in competition with pathogens. They can inhibit the pathogenic microorganism infections [7]. *Piriformospora indica* can colonize the roots of Chinese cabbage, promote the growth of roots and buds and promote the formation of lateral roots. When the transgenic plants were exposed to drought stress, the activities of peroxidase, catalase and superoxide dismutase in leaves were upregulated within 24 h, which delayed the decrease in photosynthetic efficiency and degradation of chlorophyll and thylakoid proteins caused by drought [8]. The dissolution ability of Gibberellin secreted by *Bacillus amylolyticus* and its phosphate may help promote the growth of cabbage, radish, tomato and mustard greens [9]. Therefore, mining beneficial microbial resources in nature can promote healthy plant growth and reduce the adverse effects of environmental conditions such as biological/abiotic stresses.

*Metarhizium* spp. are entomopathogenic fungi distributed worldwide, commonly existing in nature as plant rhizosphere fungi [10], endophytic fungi [11], entomopathogenic fungi [12] and soil fungi [13]. In 1883, based on previous research, Sorokin first officially named the fungus *“Metarhizum anisopliae”*. In addition to its strong insecticidal activity, has also been recognized as a biocontrol microorganism that positively affects plant pathogenic fungi [10]. Therefore, researchers have begun studying *M. anisopliae* as an entomopathogen and biopesticide. *M. anisopliae* is widely distributed in soil associated with saprophytic rhizospheres or arthropod carcasses [14]. According to previous research, *Metarhizium* spp., besides exhibiting strong insecticidal activity, has also been recognized in recent years as a biocontrol microorganism that positively affects plant growth and development and inhibits plant pathogenic fungi [13,15]. The treatment of plant seeds with microorganisms has been used to protect plants against plant pathogens and promote better plant growth. For example, *Metarhizium* spp. can change the growth of *Arabidopsis thaliana*, promote plant growth in both the above-ground part and root of *Arabidopsis* seedlings, increase the total chlorophyll content of plants, increase biomass and make plants grow more vigorously [16,17,18,19]. The vigorous growth of plants can provide a more stable habitat for *Metarhizium* spp., while as parasites, *Metarhizium* spp. can further protect the healthy growth of plants; for example, microorganisms can protect plants by dissolving inorganic nutrients or acting as biological control agents for plant pathogens. Inoculation with *Metarhizium* spp. promotes the systemic immunity of plants and confers resistance to a variety of plant pathogens [13]. For example, *Metarhizium* spp. has an inhibitory effect on *Fusarium oxysporum*, corn spot, wheat wilt, tomato wilt and strawberry gray mold. At present, the understanding of how *Metarhizium* spp. inhibit the growth of plant pathogens is still in its infancy, and the molecular mechanism of the interaction between host plants, *Metarhizium* spp. and pathogens needs to be further studied.

*Metarhizium* spp. and its secondary metabolites also have certain effects on plant hormones. By regulating the level of plant hormones, the growing conditions of crops can be improved, with results such as increases in plant height, branch number and root length; *M. anisopliae* can improve crop yield and quality or enhance crop resistance to aphids. Studies have found that *Metarhizium robertsii* or a culture filtrate containing auxin can promote the root proliferation of *A. thaliana*, activate the expression of Indole Acetic Acid (IAA) genes [20] and promote plant roots by colonizing the roots of beans, lentils, cassava, corn and switch grass, thus generating a higher initial root emergence rate and root hair density in plant roots [21,22,23]. As a result, leaf stomata become larger, the total amount of ABA and ABA metabolites is significantly reduced and there is a negative correlation between ABA changes when plants are infected by diseases and pests [11]. It was also found that the expression of immune-related genes and genes related to the synthesis of hormones such as auxin (*TaTAR23D*) and gibberellin (*TaGA20ox1*) were significantly downregulated in wheat colonization experiments, and this effect was weakened with plant growth [22,23]. *Metarhizium* spp. may delay the colonization of plant tissues by pathogens, thereby reducing disease symptoms, or *Metarhizium* spp. may alter the plant’s response to pathogens by locally activating key defense hormones in the Jasmonic acid (JA) and Salicylic acid (SA) pathways [24]. Posada-Vergara et al. used *Metarhizium* spp. to protect oilseed rape from verticillium wilt, demonstrating plant-mediated effects, and observed an improvement in plant growth and a reduction in disease symptoms, revealing the mechanism of action of *Metarhizium* spp. against soil-borne pathogens in oilseed rape and highlighting the potential of *Metarhizium* spp. as a sustainable agricultural biological control agent [25]. But the exact mechanism of how *Metarhizium* spp. promote plant growth by altering plant hormone levels is not fully understood.

*Marquandomyces marquandii* is a non-core species of *Metarhizium*, widely existing in soil. Domestic and foreign researchers have shown that the secondary metabolites of *Marquandomyces marquandii* have antibacterial and antitumor activities in the field of medical research [26]. In ecological research, this species has the potential to remove environmental pollutants (heavy metal pollution, pesticide residues, etc.), control crop pests and diseases, and promote plant growth [27,28]. However, there has been only one report of this fungus promoting plant growth [29]. In this study, the research group selected *M. marquandii* SGSF043 from forest litter and prepared a seed coating agent for Chinese cabbage. The effect of the seed coating agent with different fungus concentrations on the seed germination and bud growth of Chinese cabbage was studied in an indoor seedling tray at 24 h, 48 h, 72 h and 96 h of Chinese cabbage seed germination. At the same time, the physiological and biochemical indices, antioxidant enzyme activity and hormone levels of cabbage buds were determined 96 h after treatment, and the functional genes related to growth promotion and stress resistance were verified. This study aims to reveal that *M. marquandii* SGSF043 not only can significantly improve the germination vitality of Chinese cabbage seeds, but also has the potential to activate the plant defense mechanism in advance. The results will provide a scientific theoretical basis for further research on the mechanism by which *Metarhizium* fungi to promote the growth and disease resistance of crops.

## 2. Results

### 2.1. Effects of M. marquandii SGSF043 on Germination Vigor of Chinese Cabbage Seeds

After 48 h of germination, there were no significant differences in germination potential, germination rate and germination index between the treatment group and the control group (*p* > 0.05) (Table 1). The germination activity index data showed that there was a significant difference between the HM treatment group and the other three groups (*p* < 0.05). The bud activity index of LM was 7.70% lower than that of the control group. With the increase in the concentration in the treatment group, the bud activity index in the treatment group with MM and HM increased by 1.81% and 16.41%, respectively, compared with the control group (Figure 1A(b)).

### 2.2. Effects of M. marquandii SGSF043 on the Growth and Development of Chinese Cabbage

With the gradual growth and development of the buds (Figure 1B), at 96 h of Chinese cabbage germination, phenotypically, lateral roots of Chinese cabbage sprouts in all treatment groups showed obvious growth (Figure 1A(e),D). Compared with the control group, there was a significant difference in the above-ground portion growth in the HM treatment group (*p* < 0.05), which increased by 7.27%. The above-ground portion growth was inhibited in the LM treatment group. There was a significant difference in sprout root length among all treatment groups (*p* < 0.05), The bud root length of the HM group was the longest, followed by that of the MM group, which had a significant difference compared with the CK group (*p* < 0.05) (Figure 1C).

The results of growth indices of Chinese cabbage treated with *M. marquandii* SGSF043 coating agents showed that the seed coating agents could significantly improve the growth parameters of Chinese cabbage buds (*p* < 0.001, *p* < 0.0001). There was a significant difference in stem diameter between the three groups (*p* < 0.0001) (Appendix A). Stem diameter in the LM, MM and HM treatment groups increased by 23.42, 38.74 and 42.34% compared with the control group, respectively (Figure 2A).

As shown in Figure 2B, the MM treatment group and HM treatment group exhibited significant differences in comparison to the control group (*p* < 0.01, *p* < 0.0001), but there were no significant differences between the low-concentration treatment group and the control group. The number of secondary roots in treatment groups of fungi (LM, MM, HM) increased by 24.92, 32.16 and 53.68%, respectively, compared with the control group.

As shown in Figure 2E,F, there was a significant difference in the fresh weight of the above-ground part between the HM treatment group and the control group (*p* < 0.001). The fresh weight data of the underground part showed a significant difference between the MM treatment group and the control group (*p* < 0.1). There was a significant difference in total fresh weight between the HM treatment group and the control group (*p* < 0.0001) (Figure 2D).

By calculating the root–shoot ratio (underground dry weight/above-ground dry weight), it can be seen that there are significant differences in root–shoot ratio between the LM, MM and HM treatment groups and control group (*p* < 0.0001, *p* < 0.001, *p* < 0.0001). Compared with the control group, they increased by 118.18, 109.09 and 127.27%, respectively.

The correlation heat map of each growth index of Chinese cabbage sprouts showed that there was a significant positive correlation among all growth indices (*p* < 0.01, *p* < 0.001). Above-ground fresh weight (AFW) and underground fresh weight (UFW) were positively correlated with total fresh weight (TF) at the *p* < 0.01 level. At the *p* < 0.001 level, bud length (BH), root length (BL), stem diameter (ST), number of secondary roots (SRN) and root–shoot ratio (underground dry weight/above-ground dry weight) (RTR) were significantly positively correlated between the two groups (Figure 2G).

In Figure 2G, the darker the red color, the greater the correlation between the two groups and the larger the R value of the correlation coefficient; conversely, the lighter the red color, the weaker the correlation between the two groups and the smaller the R value of the correlation coefficient.

### 2.3. Effects of M. marquandii SGSF043 on Contents of Chlorophyll, MDA and Antioxidant Enzymes in Chinese Cabbage Buds

In Figure 3A, there was no significant difference in chlorophyll content among treatment groups at 96 h after germination (*p* > 0.05) (Figure 3A). The MDA content in the above-ground and underground parts of the buds was significantly higher in the LM treatment group than in the other three treatment groups (*p* < 0.05). The MDA content in the underground part of the sprouts increased with the increase in the concentration of seed coating agent fungus, and the MDA content in the LM treatment group was the lowest in the roots of the buds (Figure 3B). The content of MDA in the above-ground part was negatively correlated with that in the underground part.

The results of antioxidase data of Chinese cabbage showed that the use of different fungus concentrations of coating agents had certain effects on the antioxidase of Chinese cabbage bud tissue. The results of superoxide dismutase (SOD) can be seen (Figure 3C), The enzymes activity measurement data of the underground part of the buds showed that there were significant differences between the LM treatment group, the HM treatment group and the CK group (*p* < 0.05). In the peroxidase (POD) treatment groups (Figure 3D), the above-ground data showed that the LM treatment group and control group exhibited significant differences in comparison to the MM and HM treatment groups (*p* < 0.05). The LM treatment group and HM treatment group exhibited significant differences in comparison to the control group (*p* < 0.05). In Figure 3E, it can be seen that the MM treatment group had the most significant effect on catalase (CAT) in various tissue parts of Chinese cabbage buds (*p* < 0.05). With the increase in fungus concentration, LM and HM treatment had an inhibitory effect on the catalase of young Chinese cabbage leaves and decreased the catalase activety of young Chinese cabbage leaves compared with the control group. MM treatment increased catalase activity in young bud roots (*p* < 0.05).

### 2.4. Effect of M. marquandii SGSF043 on the Content of Hormones in Chinese Cabbage

According to the analysis of the measurement data of hormone content in the young buds for 96 h, the addition of fungus can affect five kinds of hormones, namely spermidine, GA, Jasmonic acid (JA), Indole Acetic Acid (IAA) and ABA, in the above-ground part of the young buds (Figure 4). The contents of GA and Jasmonic acid (JA) in underground roots of young buds were increased by HM treatment. As shown in Figure 4A, there was a significant difference between the above-ground spermidine of the sprouts in the LM treatment group and that in the other three groups (*p* < 0.05). Compared with the control group, the spermidine content in the roots of the sprouts in the treatment group had no effect. As shown in Figure 4B, the GA content in the above-ground part of the buds gradually increased with the increase in fungus concentration. There were significant differences between the MM and HM treatment groups and the control group (*p* < 0.05). The contents of GA in the roots of young buds were affected by the HM treatment group. As can be seen from the determination results of Jasmonic acid (JA) content (Figure 4C), there was a significant difference in the hormone content of sprout roots between the HM treatment group and the control group (*p* < 0.05). IAA and ABA data showed that with the gradual increase in the concentration of seed coating agent treatment, the hormone content in the above-ground part of the aerial parts was increased, but the corresponding hormone content in the root system was not affected. The bud Indole Acetic Acid (IAA) data showed a significant difference between the HM treatment group and the control group (*p* < 0.05) (Figure 4D). The ABA data showed that there was a significant difference between the HM treatment group and the LM and control groups (*p* < 0.05) (Figure 4E).

### 2.5. Expression of Genes Related to Growth Promotion and Resistance in Chinese Cabbage

Based on the observation of seedling phenotype, growth and development indicators, and physiological and biochemical indicators of Chinese cabbage, this part of the experiment analyzed the differences in the expression of functional genes related to growth and stress resistance of Chinese cabbage. We screened 10 genes, namely *BrPAO, ARF7, ARF19, BrSAG12, BnNCED3, BrSOD, BrPOD, BrCAT, MYC3* and *IAA*. We performed qRT-PCR verification on 96 h bud samples from the CK group and HM treatment group (Figure 5).

As can be seen from the chlorophyll physiological parameter diagram, there was no significant difference in chlorophyll content between the HM treatment group and the control group. Compared with the control group, the relative expression level of the chlorophyll-related gene *BrPAO* in the leaves and roots of the treatment group was significantly different (*p* < 0.05, *p* < 0.0001). The transcription level in leaves was upregulated, and the relative expression level was 2.68 times, while the expression level in roots was downregulated, with only a small expression level. The *ARF7* and *ARF19* genes related to lateral root germination in Chinese cabbage can regulate lateral root germination in Chinese cabbage. Compared with the CK group, the relative transcription levels of the *ARF7* and *ARF19* genes in HM-treated Chinese cabbage were significantly increased (*p* < 0.001, *p* < 0.01). The relative expression levels in the HM group were 5.23 and 9.92 times, respectively. Compared with the control group, the *BrSAG12* and *BnNCED3* genes related to ABA synthesis were not expressed in the leaves of the HM treatment group, while the transcription level of the *BnNCED3* gene in the leaves was significantly increased (*p* < 0.0001), and the relative expression level was 1.62 times. The transcription levels of the *BrSAG12* and *BnNCED3* genes are relatively low in roots. The relative expression levels of the superoxide dismutase-related gene *BrSOD*, peroxidase-related gene *BrPOD* and catalase-related gene *BrCAT* in the above-ground and underground parts of Chinese cabbage were analyzed, and the transcription levels of the *BrSOD* and *BrPOD* genes in leaves of the HM treatment group were significantly increased compared with control group (*p* < 0.05). The relative expression levels of the *BrCAT* gene were 1.44 times and 1.27 times, respectively. The transcription levels of the *BrCAT* gene were significantly decreased (*p* < 0.0001), and the relative expression levels were low. The relative expression levels and transcription levels of the *BrSOD*, *BrPOD* and *BrCAT* genes in the underground part of Chinese cabbage were significantly downregulated (*p* < 0.0001). The high fungal concentration inhibited the expression of antioxidase genes in the roots of Chinese cabbage buds. The *IAA* gene in the roots was significantly lower than that in the control group, the transcription level of the *IAA* gene in leaves was significantly higher than that in the control group, and the relative expression level was 1.79 times. Compared with the control group, the *MYC3* gene of MYC, the core regulator of the Jasmonic acid (JA) signaling pathway, was almost not expressed in the leaves after fungal treatment, and the transcription level of the roots was significantly downregulated (*p* < 0.0001). Compared with the control group, the gene transcription level in the above-ground part of Chinese cabbage sprouts was generally significantly upregulated, and the related gene transcription level in the underground part of Chinese cabbage buds was generally downregulated. The results showed that the fungal treatment affected the transcription levels of related regulatory genes in different tissue parts of Chinese cabbage buds.

### 2.6. Principal Component Analysis (PCA)

Principal component analysis (PCA) was used to analyze the biomass, chlorophyll, MDA, antioxidant enzymes, plant hormones and parameters of each part of 96 h Chinese cabbage buds. The x- and y-axes represent the first (PC1) and second (PC2) principal components, respectively (Figure 6), with a total PCA interpretation of about 83.736%. PC1 is the treatment difference among different fungus concentration levels, and PC2 is the effect of each treatment group on Chinese cabbage buds. The results of this study showed that with the increase in the concentration of fungus coating agent, the HM treatment group and the control group were obviously separated. The higher the concentration of fungus, the more obvious the effect. The lower the concentration of fungus, the smaller the dispersion from the control group.

## 3. Discussion

### 3.1. Effects of M. marquandii SGSF043 on the Growth and Development of Chinese Cabbage Buds and Antioxidant Enzyme Indices

In order to study the interaction between plants and microorganisms and its influence on plant growth and development, in this experiment, we used *M. marquandii* SGSF043 isolated in the laboratory to prepare seed coating agents with different fungal spore contents to treat Chinese cabbage seeds. The results showed that in the initial stage of seed germination, the germination potential, germination rate and germination index of Chinese cabbage seeds were not significantly improved by the seed coating agent treatment, which indicated that water was one of the key factors affecting the germination of Chinese cabbage seeds. With the extension of germination time, especially at 96 h, we observed that the lateral roots of the buds in the treatment group increased significantly. This is consistent with the results of previous studies on the promotion of lateral root increase in maize buds by *M. marquandii* SGSF043 [19]. In addition, we found that there was no significant difference in leaves chlorophyll content among the groups treated with the low-concentration seed coating agent at 96 h, which was different from the results of previous studies on growth-promoting fungi that can increase plant chlorophyll content, possibly because the development time of young buds was shorter, the leaves were smaller and the photosynthetic efficiency was not as high as that of mature leaves [30]. The LM treatment group can reduce the content of MDA in roots, which may indicate that the low-concentration treatment may activate the growth and metabolic pathways of plants and improve the antioxidant capacity of plants, while the high-concentration treatment may activate the defense and detoxification pathways of plants, reduce the content of MDA and alleviate oxidative stress [31]. Studies have shown that *Metarhizium* strain MetA1 (MA) plays a role in promoting the salt tolerance of rice. It can increase the activity of antioxidant enzymes and improve the salt tolerance and yield of plants by decreasing the levels of MDA and hydrogen peroxide. These results indicate that physiological and biochemical mechanisms of plants can be regulated according to different concentrations of fungi, enhancing plant tolerance to environmental stress [32,33].

Studies have shown that *Metarhizium* mostly resides in plant roots, so the roots, as the organs of plants in direct contact with external environmental microorganisms, are more sensitive to changes in *Metarhizium* spp. concentration, and leaves respond indirectly to root signals through other signal transduction pathways, thus showing different changes in antioxidant enzyme activity. In a study of the effects of *Metarhizium* spp. on insects, it was found that *Metarhizium* spp. conidia could affect the activity of physiological and biochemical indices, including a change in antioxidant enzyme levels, while carrying out biological control on *Spodoptera littoralis* [34]. The results showed that *M. marquandii* SGSF043 had a more direct effect on the roots of plants, while the leaves showed different antioxidant activity in response to root signals through signal transduction pathways. These findings provide a new direction for further research on the mechanism of *Metarhizium* spp. on plants under different environmental conditions and provide a scientific basis for using *Metarhizium* spp. to promote crop growth and improve stress resistance.

### 3.2. Effects of M. marquandii SGSF043 on Chinese Cabbage Bud Hormones

The interaction and balance between plant hormones are essential for the normal growth of plants, and external factors, including microorganisms that live symbiotically with plants, can have a significant impact on certain hormone levels in plants. Some microbial metabolites can also directly or indirectly promote or inhibit the level of plant hormones when co-cultured with plants, such as cyclic liptides (Metarrhizin Destruxins, DTXs), metabolites of *Metarhizium* spp. Studies have found that a large number of DTXs were detected in the co-culture of four kinds of *Metarhizium* spp. with soybeans and corn, including 25 DTX analogs. DTX production varies significantly under different plant conditions, and the functional definition of these metabolites is still incomplete. Studies have also shown that there may be a chemical signal exchange mechanism between plant hormones released by plants and *Metarhizium* spp., thus contributing to the establishment of fungus–plant symbiosis [22,35,36,37,38]. Studies have also shown that *Metarhizium* spp. can activate the plant immune system by secreting growth hormones (indole-3-acetic acid, tryptamine, indole-3-acetamidotryptophan, piperidinic acid) that are essential for plant development and improve plant defense against different stress conditions [39]. After the analysis of the results of the determination of the content of germination hormone after 96 h, it was found that the HM treatment group had the greatest influence on the content of Jasmonic acid (JA) in roots compared with the control group. The effect of LM on the content of spermidine was the greatest, the effect of MM and HM on the content of GA was the greatest, and the effect of HM on leaf Indole Acetic Acid (IAA) and ABA was the greatest. Different concentrations of seed coating agents had different degrees of influence on budding hormones. This may be related to the types of metabolic secretions of *Metarhizium anisopliae* and the content of corresponding compounds. At present, further detailed studies are needed on the specific mechanism of the metabolites of *M. marquandii* and how they affect plant hormones when acting on plants.

### 3.3. Expression of Related Functional Genes

By staining the roots of Chinese cabbage, the presence of *M. marquandii* was not detected, but the qRT-PCR results showed that the gene transcription level of the strain was expressed in the roots. In order to further clarify the effect of *M. marquandii* on improving seed germination vitality, combined with the phenotypic characteristics and physiological and biochemical indicators of Chinese cabbage, we conducted real-time fluorescence quantitative qRT-PCR verification of functional genes related to the growth and stress resistance of Chinese cabbage. After fungus treatment, the transcription levels of genes in the above-ground part of Chinese cabbage buds were generally significantly upregulated, while the transcription levels of related genes in the underground part of Chinese cabbage buds were generally downregulated. The fungus affected the transcription levels of related regulatory genes in different tissue parts of Chinese cabbage buds.

#### 3.3.1. Regulation of Genes Related to Chlorophyll Synthesis and Lateral Root Development by *M. marquandii*

Previous studies have reported that after the application of *Metarhizium* spp., the chlorophyll content in leaves will first increase and then decrease during seed germination to seedling maturity, which may be related to the activation of the *BrPAO* gene, promoting the upregulation of chlorophyll catabolism genes and reducing chlorophyll content. This also indicates that there is a mechanism of resource redistribution from old leaves to new leaves and growth points in plants. At 96 h after the germination of Chinese cabbage seeds, *M. marquandii* was mainly involved in the upregulation of the transcription level of the gene *BrPAO*, which is related to chlorophyll degradation during leaf aging [40]. However, the downregulation of *BrPAO* expression in roots may be related to the way plants regulate their own immune response during plant interaction [41].

In the early stage of plant growth and development, two auxin response factor family members, *ARF7* and *ARF19*, can activate the expression of *IAA* response *LBD/ASL* genes and mainly participate in the regulation of lateral root formation [42]. *ARF7 and ARF19* are located in the nucleus of the plant and relocated in the differentiation region of the root [43]. E3 ubiquitin ligase SCFAFF1 regulates the accumulation and distribution sites of *ARF7* and *ARF19* [44]. Fang et al. [45] found that plant hormones, environmental factors and non-coding RNA play important roles in regulating *ARF* function. Thus, the regulation of the *ARF* gene affects the formation and development of plant roots, stems, leaves and other organs. In this experiment, the *ARF7* and *ARF19* genes were preliminarily analyzed by qRT-PCR. The results showed that the overexpression of *ARF7* and *ARF19* could be induced by *M. marquandii*, which had the potential to improve nutrient uptake and transport in the roots of Chinese cabbage, which was consistent with the results of the study on sprout phenotype and growth and development in this paper. However, to determine whether promoting the growth of lateral roots can improve the inhibitory effect of plants on pathogenic fungi/bacteria, the regulatory mechanism of *ARF* in the mechanism of plant disease resistance can be further studied.

#### 3.3.2. Regulation of Functional Genes Related to the Regulation of Antioxidation System in *M. marquandii*

*Metarhizium* treatment may have triggered antioxidant systems in the leaves, enhancing the plant’s tolerance to biological and abiotic stresses [46]. Superoxide dismutase (SOD) and peroxidase (POD) are key enzymes in plant response to oxidative stress, and they play an important role in clearing reactive oxygen species (ROS) [46,47]. Studies have shown the effects of biocontrol strain B1619 on the activity of peroxidase (POD) and superoxide dismutase (SOD) in tomato plants, thus proving that biocontrol strain B1619 can induce the expression of disease-resistant genes in tomato leaves and enhance the disease resistance of plants. By fluorescence quantitative qRT-PCR, it was found that the expression levels of *pod1* related to SOD enzyme synthesis and *sod*, a related gene, were upregulated after B1619 treatment and reached the highest value at 48 h, which was consistent with the results of this experiment [48]. The results of this study showed that compared with the control group, the expression of the *BrSOD* and *BrPOD* genes in the leaves of Chinese cabbage in the treatment group was significantly upregulated, while the expression in the roots was significantly downregulated. Previous literature focused on the effects of biological and abiotic stresses on the antioxidant system of plant leaves, and studies on the mechanism of plant root diseases could be carried out in combination with existing studies. The downregulation of the *BrSOD* and *BrPOD* genes in roots may indicate that the effects of *M. marquandii* on roots are different from those on leaves. This difference may be related to the different roles of roots and leaves in plant physiology and defense mechanisms. The roots, as the organs of the plant in direct contact with the soil, may be more sensitive to the response of *Metarhizium* spp., resulting in downregulated expression of its antioxidant enzyme genes to adapt to specific environmental conditions. Catalase may be involved in breaking down hydrogen peroxide to reduce oxidative damage in plants. *CAT* gene expression may be downregulated during plant resistance to disease invasion as part of the antioxidant defense system, which is normally responsible for removing hydrogen peroxide to protect plants from oxidative damage. However, by activating the immune response of plants and inducing systemic resistance (SAR), moderate levels of hydrogen peroxide may be beneficial to plant growth resistance [49]. The results of this study showed that the expression of this gene was consistent with that of *BrCAT*, and the gene expression was downregulated in the leaves and roots of Chinese cabbage sprouts, which may reflect the regulation of the antioxidant system of Chinese cabbage buds by *M. marquandii*, so that the plants can maintain an appropriate reoxidation state in response to environmental changes and help the plants balance ROS levels. This may be related to the effect of *Metarhizium* spp. on plant metabolic pathways. However, the direct regulation mechanism of plant antioxidant system by biocontrol and growth-promoting fungi has not been reported.

#### 3.3.3. Regulation of Functional Genes of Plant Immunity and Disease Resistance

The expression of the *BrSAG12* gene is regulated by ABA and Jasmonic acid (JA), two plant hormones that promote aging. Exogenous application of methyl JA (MeJA) can promote the leaf aging of Chinese cabbage, and MeJA treatment can reduce the maximum mass seed yield (Fv/Fm), photosynthetic electron transfer rate (ETR) and total photosynthetic intensity of plants. At the same time, the expression of several aging-related genes (SAGs) such as *BrSAG12* and the chlorophyll catabolism gene *BrPAO1* was significantly induced [50,51]. In the aging process, plants treated with MeJA can eliminate phototoxic chlorophyll precursors and activate enzymatic antioxidant reactions to form a protective mechanism to reduce oxidative stress. Song Yun’s research group found that the growth of Jasmonic acid (JA) on taproot depends on the Indole Acetic Acid (IAA) signaling pathway, while the growth of lateral roots depends on *ARF7*, *ARF19* and other genes [52].

Overexpression of *BnNCED3* contributes to ABA accumulation and the production of nitric oxide (NO) and reactive oxygen species (ROS), thereby increasing tolerance to abiotic stresses such as drought and salt stress. Transcriptional analysis showed that the expression level of the *BnNCED3* gene was different in different tissue parts of brassica napus, especially in senescent leaves [53], The overexpression of this gene was involved in the inhibition of seed germination, initiation of lateral roots and enhancement of ABA-related leaf senescence. The results of this experiment showed that the expression level of the *BnNCED3* gene in leaves treated with a fungus coating agent was higher than that in the control group. According to the results of the seed germination experiment, the seed germination of Chinese cabbage in all treatment groups was not inhibited by this gene (Table 1). Whether the mechanism of lateral root initiation of Chinese cabbage in the experimental treatment group is related to this gene needs to be further verified in environmental stress resistance tests.

During plant growth and development, the *MYC3* gene may be involved in various hormone signal interactions, jointly affecting plant growth and development and stress response, including Abscisic Acid (ABA), gibberellin (GA), etc. *MYC3* is involved in regulating the response of plants to Jasmonic acid (JA) signals and activating Jasmonic acid (JA) response genes in response to trauma, pathogen invasion and insect biting [54]. In the determination of the physiological indices of Jasmonic acid (JA) in plants, the Jasmonic acid (JA) content of each treatment group was different in the roots at 96 h, and the Jasmonic acid (JA) content was the highest in the roots treated with HM, which may be due to the fact that the biosynthesis and signal transduction pathway of Jasmonic acid (JA) are regulated by various factors.

At the molecular level, Indole Acetic Acid (IAA) synthesis and signaling are regulated by a complex network of genes. Indole Acetic Acid (IAA) plays a crucial role in the interaction between plants and growth-promoting bacteria [55]. Studies by Liao et al. have shown that *Metarhizium Robert* possesses a tryptamine (TAM)- and indole-3-acetate amine tryptophan (Trp)-dependent auxin biosynthesis pathway. IAA production in the *Metarhizium Robert* strain with *Mrtdc* knocked out was blocked because the conversion of tryptophan to TAM was blocked, but the deletion of this gene did not affect the strain’s growth-enhancing properties. At the same time, it has also been reported that Indole Acetic Acid (IAA) produced by microorganisms affects root structure, nutrient uptake and resistance to abiotic stresses, thereby enhancing plant adaptability [56]. The results of this study showed that the fungal seed coating agent could significantly increase the Indole Acetic Acid (IAA) content in the leaves of Chinese cabbage seedlings, and at the same time, the Indole Acetic Acid (IAA) transcription level of the genes related to Indole Acetic Acid (IAA) synthesis in the leaves was upregulated, suggesting that the fungal seed coating agent may have a mechanism of promoting plant growth by increasing the Indole Acetic Acid (IAA) content in plant leaves [57]. It is speculated that the Indole Acetic Acid (IAA) biosynthesis pathway in Chinese cabbage buds may be activated by secreting specific signaling molecules. The plant immune defense system is partially activated by *M. marquandii*; the specific signal molecules need to be further studied.

## 4. Materials and Methods

### 4.1. Preparing Materials

Test plant: Cabbage variety “Beijing New No. 3”. Seeds were purchased from seed markets, Shandong Shouhe, Shouguang Xinranran Horticulture Co., Ltd. (Shouguang, China).

Test strain: *M. marquandii* SGSF043. It is preserved by the Key Laboratory of Ecological Restoration and Resource Utilization of Heilongjiang University (Harbin, China).

Isolation and molecular identification of strains: *M. marquandii* SGSF043 was isolated from the litter in the Nanwenghe National Nature Reserve in Daxing’anling region, Heilongjiang Province. The main plant species were *Tilia amurensis* Rupr., *Quercus mongolica* Fisch ex Ledebour, *Fraxinus mandshurica* Rupr., etc. The litter was divided into the undecomposed layer, semi-decomposed layer and decomposed layer from top to bottom. After collecting the samples in layers, we brought them back to the laboratory and placed them in a cool place for natural air drying for separation and purification. The DNA of the fungal strain was extracted by the CTAB method and the internal transcribed spacer (ITS), known for its use in initial identification, was amplified with the universal primers by ITS1/ITS4 for fungi [58,59]. According to Bischoff et al., primers EF-983F (5′—GCYCCYGGHCAYCGTGAYTTYAT-3′) and EF-2218R (5′—ATGACACCRACRGCRACRGTYTG—3′) were used to further amplify the sequence fragment of translation extension factor 1 α (translation elongation factor 1 α, *TEF*) of the tested strain. The PCR products were sequenced bidirectionally and analyzed by BLAST from the NCBI website [60,61]. The above strain has been preserved in the strain collection center of Wuhan University, CCTCC No. M2020555.

Preparation of spore suspension: Equal amounts of pre-activated strain *M. marquandii* SGSF043 were transferred into sterile Erlenmeyer flasks, each containing 100 mL of Potato Dextrose Broth (PDB) liquid medium. The inoculated flasks were placed in a constant-temperature shaker incubator at 25 °C, with a shaking speed of 180 rpm^−1^, and cultured in the dark for 14 d; after 14 d of cultivation, the fungal mycelium in the fermentation broth was filtered out using sterile filter paper to obtain the spore-containing fermentation broth; the concentration of viable fungal spores in the spore fermentation broth was adjusted to three specific concentrations: 1 × 10^6^, 1 × 10^7^ and 1 × 10^8^ colony-forming units per milliliter (CFU/mL). A standard hemocytometer was used for the adjustment to ensure accurate quantification.

### 4.2. Preparation of Chinese Cabbage Seed Coating Agent

Seed disinfection and preparation: Cabbage seeds were subjected to a disinfection process prior to use. Initially, the seeds were washed with sterile water. Subsequently, they were soaked in a solution containing 70% anhydrous ethanol and 25% sodium hypochlorite for 1 min, respectively. After the soaking treatment, the seeds were rinsed with sterile water three times to ensure complete removal of the disinfecting agents. Formulation of seed coating agents: for the treatment group, a seed coating agent was prepared by incorporating a specific quantity of acacia gum into spore suspensions of varying concentrations. This mixture was then adjusted to create a 10% acacia gum seed coating agent using the supernatant of the fermentation liquid (Figure 7①②③). In contrast, the control group’s seed coating agent was prepared using only the PDB liquid medium, resulting in a 10% acacia gum seed coating agent. Seed coating process: Pre-sterilized cabbage seeds were thoroughly mixed into the pre-prepared seed coating agents to ensure that every part of the seeds was uniformly coated with the solution. The spore content for each treatment group was standardized, and the coated seeds were allowed to dry on a super-clean table (Figure 7④⑤).

### 4.3. Experimental Design

Experimental design and coating agent preparation: In this experiment, a total of four seed coating agents were established. These included a low concentration (LM) with an effective viable fungal count of 1 × 10^6^ CFU/mL, a medium concentration (MM) with a count of 1 × 10^7^ CFU/mL, a high concentration (HM) with a count of 1 × 10^8^ CFU/mL and a blank control (CK) treated directly with PDB medium. The average spore content per seed in the three fungal treatment groups was determined to be 5.6 × 10^4^, 5.6 × 10^5^ and 5.6 × 10^6^ CFU/mL, respectively, based on the average spore content of 10 seeds. Germination experiment setup: A seedling tray, measuring 32.5 cm in length, 24.5 cm in width and 4.5 cm in height, was selected and soaked in 75% alcohol to serve as the container for the Chinese cabbage seed germination experiment. Sterile gauze was laid out in the seedling tray, and 50 mL of sterilized water was poured to wet the gauze. The cabbage seeds, coated with the respective seed coating agents, were then evenly distributed into the seedling tray, with approximately 500 seeds per tray. Each treatment was set up in triplicate, resulting in a total of 12 trays. The prepared seedling trays for each treatment group were incubated at a room temperature of 25 °C with a photoperiod of 16 h light to 8 h dark (Figure 7⑤).

### 4.4. Effects of M. marquandii SGSF043 on Seed Germination of Chinese Cabbage

The seed coat of cabbage in each treatment group cracked after about 6 h (Figure 1A(a–e). The germination of Chinese cabbage seeds at 48 h was recorded from the beginning of germ leakage 24 h (Figure 1A(a–e), and the germination potential (%), germination rate (%), germination index and bud activity index were calculated.
(1)$$\mathrm{Germination}\;\mathrm{potential}\;(\%)\;\mathrm{GP}/\%=A_{2d}/A_{t}\times 100$$

In this formula, GP stands for the germination potential (%), *A_2d_* is the number of germinated seeds after 48 h and *A_t_* is the total number of experimental seeds.
(2)$$\mathrm{Germination}\;\mathrm{rate}\;(\%)\;\mathrm{GE}/\%=A_{c}/A_{t}\times 100$$

In this formula, GE is the germination rate %, *A_c_* is the number of germinated seeds at 48 h after germination and *A_t_* is the total number of experimental seeds.
(3)$$\mathrm{Germination}\;\mathrm{index}\;\mathrm{GI}=(\sum G_{t}D_{t})$$

Here, GI is the germination index, *G_t_* is the germination number after *“t”* days, *D_t_* is the corresponding number of germination days and *“t”* is 48 h (2d) [28].
(4)Buds activity index VI=GI×S

Here, VI is the bud vitality index, GI is the germination index and *S* is the bud root length.

### 4.5. Effects of M. marquandii SGSF043 on the Growth and Development of Chinese Cabbage Buds

Measurements of Chinese cabbage bud growth and development: To further ascertain the impact of the seed coating agent on the growth and development of Chinese cabbage buds, a series of measurements were conducted on the buds 96 h post-seeding. The bud root length, total bud length, stem diameter number of secondary roots, fresh weight above ground, fresh weight below ground, and total fresh weight of Chinese cabbage buds were determined using a vernier caliper. Additionally, the root–shoot ratio, calculated as the ratio of underground dry weight to above-ground dry weight, was also ascertained [62].

### 4.6. Determination of Chlorophyll, MDA and Antioxidant Enzymes in Leaves of Chinese Cabbage Buds

**Determination of total chlorophyll content:** The total chlorophyll content was determined using 95% anhydrous ethanol. Young leaves of Chinese cabbage grown for 96 h were sampled, with 0.1 g of the leaves being taken and a small amount of quartz sand added. Subsequently, 1 mL of 95% anhydrous ethanol was added, and the mixture was ground until the tissue appeared white. The ground mixture was allowed to stand for 5 min before the solution was transferred to a 0.45 µm water filter. The absorbance of the filtrate was measured at wavelengths of 665 nm and 649 nm, and the total chlorophyll content was calculated based on the absorbance data at these different wavelengths using a specified formula. Each treatment group was repeated three times to ensure the accuracy and reliability of the results [62].
(5)$$C_{a}\;(\mathrm{mg}/L)=13.95A_{665}-6.88A_{649}$$


(6)
$$C_{b}\;(\mathrm{mg}/L)=24.96A_{649}-7.32A_{665}$$



(7)
$$C_{c}=+C_{b}$$



(8)
$$C\;(\mathrm{mg}/L)=\frac{C_{c}\times V\times N}{W}$$


Here, C_a_ is the chlorophyll A concentration (mg/L), C_b_ is the chlorophyll B concentration (mg/L), C_C_ is the total chlorophyll concentration (mg/L), C is the total chlorophyll content (mg/g), *V* is the extraction liquid volume (mL), *N* is the dilution ratio and *W* is the sample fresh quantity (g).

Malondialdehyde (MDA) content determination: Cabbage bud samples, comprising leaves and roots, were collected for the measurement of malondialdehyde (MDA) content. Specifically, 0.1 g of the sample was taken, and 1 mL of a 5% trichloroacetic acid solution was added. The sample was ground and then centrifuged at 3000 rpm for 20 min at 4 °C, after which the supernatant was harvested as the crude extract [30]. A mixture was prepared by combining 0.5 mL of the crude extract with 0.5 mL of a thiobarbituric acid solution containing 0.67% (*W*/*V*). This mixture was incubated at 100 °C for 30 min and subsequently cooled completely on ice. Following centrifugation at 3000 rpm for 10 min at 4 °C, the absorbance of the supernatant was recorded at wavelengths of 450 nm, 532 nm and 600 nm. The MDA concentration was determined using a specific formula, which was then used to calculate the MDA content in the plants. Each treatment group was repeated three times to ensure the precision of the measurements [62].
(9)$$C\;(\mu \mathrm{mol}/L)=6.45(A_{532}-A_{600})-0.56A_{450}$$


(10)
$$Y\;(\mu \mathrm{mol}/g)=\frac{CV}{W}$$


In this formula, C is the MDA concentration (μmol/L), *V* is the extraction liquid volume (mL), *W* is the plant tissue fresh weight (g) and *Y* is the MDA content (μmol/g).

Determination of catalase (CAT) activity: The activity of catalase (CAT) was assessed using potassium permanganate titration. Samples consisting of 0.2 g of both above-ground and underground parts of the cabbage were collected and mixed with 0.2 mol/L pH 7.8 phosphate buffer. The mixture was ground into a homogenate and transferred to a 10 mL volume bottle, and the mortar was rinsed repeatedly with the buffer to ensure the volume was accounted for. The homogenate was centrifuged at 4000 rpm for 15 min, and the supernatant was collected as the crude extract of catalase. A 50 mL triangular bottle was used as the test vessel, into which 0.25 mL of enzyme liquid was added. An equal amount of dead enzyme liquid was added to the control bottle, and 0.25 mL of 0.1 mol/L H_2_O_2_ was added to both. The bottles were maintained in a constant-temperature water bath at 30 °C for 10 min, after which 0.25 mL of 10% H_2_SO_4_ was added immediately. The solution was then titrated with a 0.1 mol/L KMnO_4_ standard solution (previously calibrated with a 0.1 mol/L oxalic acid solution) until a persistent pink color appeared, which did not fade within 30 s, indicating the end point of the titration. Each treatment group was repeated three times to ensure the accuracy and reliability of the results [62].
(11)$$\mathrm{CAT}\;(\mathrm{mg}/g)=\frac{(A-B)\times \frac{V_{T}}{V_{S}}\times 1.7}{W\times t}$$

In this formula, *A* is the amount of KMnO_4_ in contrast titration (mL), *B* is the amount of KMnO_4_ titrated after enzyme reaction (mL), *V_T_* is the total amount of extracted enzyme solution (mL), *W* is the sample fresh quantity (g), *t* is the reaction time (min) and 1.7 indicates that 0.1 mol/L KMnO_4_ 1 mL is equivalent to 1.7 mg H_2_O_2_.

Determination of superoxide dismutase (SOD) activity: The activity of superoxide dismutase (SOD) was determined using the nitrogen blue tetrazolium (NBT) method. Samples consisting of 0.1 g of the aerial and underground tissues of the cabbage were collected, and 0.2 mL of 0.05 mol/L phosphate buffer solution was added under ice-cold conditions. The tissues were homogenized, and the homogenate was diluted to 5 mL with the buffer solution. The mixture was then centrifuged at 4000 rpm for 10 min, and the supernatant, which was the crude enzyme extract, was collected. Transparent cuvettes were prepared with 1.5 mL of 0.05 mol/L phosphate buffer solution, 0.3 mL of 130 mmol/L Met solution, 0.3 mL of 750 μmol/L NBT solution, 0.3 mL of 100 μmol/L EDTA-Na_2_ solution, 0.3 mL of 20 μmol/L riboflavin, 0.05 mL of enzyme solution and 0.25 mL of distilled water, bringing the total volume to 3 mL per tube. A blank control group was established by replacing the enzyme solution with a buffer solution in the control group. After thorough mixing, one control tube was kept in the dark, while the other tubes were exposed to 4000 Lx sunlight for 20 min. Once the reaction was complete, the control tube not exposed to light was used as a blank, and the absorbance of each of the other tubes was measured at 560 nm. Each treatment group was repeated three times to ensure the precision of the measurements [62].
(12)$$\mathrm{SOD}\;(U/g)=\frac{{(A}_{ck}-A_{E})\times V}{\frac{1}{2}\times A_{ck}\times W\times V_{t}}$$

In this formula, SOD is the total SOD activity (U/g), *A_ck_* is the absorbance of the lighting charge, *A_E_* is the absorbance of the sample tube, *V* is the total volume of sample liquid (mL), *V_t_* is the sample amount (mL) at the time of determination and *W* is the sample fresh quantity (g).

Determination of peroxidase (POD) activity: The activity of peroxidase (POD) was determined using the guaiacol method. Tissue samples, consisting of 0.1 g from both the aerial and underground parts of cabbage, were collected, and 2 mL of 0.05 mol/L pH 5.5 phosphate buffer solution was added. The mixture was ground into a homogenate and then centrifuged at 3000 rpm for 10 min. The supernatant, which served as the crude enzyme extract, was collected. The enzyme activity assay reaction system was prepared, consisting of 2.9 mL of 0.05 mol/L phosphate buffer solution, 1.0 mL of 2% H_2_O_2_, 1.0 mL of 0.05 mol/L guaiacol and 0.1 mL of enzyme solution. As control, POD enzyme extract was boiled for 5 min in each treatment group. Upon the addition of the enzyme solution to the reaction system, the mixture was immediately incubated in a water bath at 37 °C for 15 min and then quickly transferred to an ice bath. To terminate the reaction, 2.0 mL of 20% trichloroacetic acid was added, and the mixture was centrifuged at 5000 rpm for 10 min. Finally, the absorbance was measured at a wavelength of 470 nm. Each treatment group was repeated three times to ensure the accuracy and reproducibility of the results [62].
(13)$$\mathrm{POD}\;(U/g)=\frac{{∆A}_{470}\times V_{T}}{W\times V_{s}\times 0.01\times t}$$

In this formula, Δ*A_470_* is the absorbance change during the reaction time, *W* is the sample fresh quantity (g), *t* is the reaction time (min), *V_T_* is the total volume of extracted enzyme solution (mL) and *V_s_* is the enzyme liquid volume (mL) used for determination.

### 4.7. Determination of Hormone Content in Chinese Cabbage Buds

Following the manufacturer’s guidelines, we employed the following products from Beijing Boaotoda Technology Co., LTD., based in Beijing, China, to determine the levels of specific hormones in 96 h Chinese cabbage buds (leaves, roots): Phytospermine (Product Code: TOPEL30097); Gibberellic Acid (GA) (Product Code: TOPEL03457); Jasmonic acid (JA) (Product Code: TOPEL30282); Indole Acetic Acid (IAA) (Product Code: TOPEL30282); Abscisic Acid (ABA) (Product Code: TOPEL03473)

### 4.8. Extraction of mRNA and Reverse Transcription of cDNA from Chinese Cabbage Buds

The extraction and reverse transcription of Chinese cabbage cDNA mRNA were studied by grinding fresh Chinese cabbage sprouts and roots with liquid nitrogen and conducting histopathological division with TRIzol reagent. The extracted RNA was verified for completeness and purity using the NanoDrop 2000c system (Thermo Scientific, Pittsburgh, PA, United States). Reverse transcription was then performed with TAKARA RR036A PrimeScript™RT Master Mix (TAKARA, Japan). For specific experimental procedures, please refer to the TAKARA Total RNA Isolation and Extraction Manual and the TAKARA RR036A PrimeScript™RT Master Mix reverse transcription instructions.

### 4.9. Real-Time Fluorescence Quantitative qRT-PCR Test

The primer sequence of related genes was quoted from previous research results (Appendix A), and the primer was synthesized by Kumei Biological Company (Changchun, China). The reaction system was configured according to the template tracer dye quantitative PCR assay kit (TB Green^®^Premix Ex TaqTM, TAKARA). Each real-time fluorescent quantitative PCR mixture consisted of 15 μL 2 × TB GreenPremix Ex Taq (2I), 1.2 μL primer, 3 μL ROX Reference Dye (50I), cDNA diluted to 500 ng and 9 μL ddH_2_O. The total reaction system was 30 μL. qRT-PCR was performed using a REFA 40425 fluorescent quantitative PCR device (Cottage Technologies Holdings Ltd., Thermo Fisher Scientific, UK). The reaction conditions are selected according to the manufacturer’s instructions and the temperature required for the relevant primers. The relative expression levels of transcription samples were analyzed by 2^−ΔΔct^. Each process was repeated 3 times. Ten functional genes related to growth promotion and stress resistance of Chinese cabbage were selected, namely the chlorophyll-related *BrPAO* gene, the *AFR7* gene and *AFR19* gene of Chinese cabbage related to the promotion of lateral root growth, the *BrSAG12* gene and *BnNCED3* gene related to ABA synthesis, the superoxide dismutase-related *BrSOD* gene, the peroxidase-related *BrPOD* gene, the catalase-related *BrCAT* gene, the transcription factor MYC family *MYC3* gene involved in the Jasmonic acid (JA) signaling pathway, and the auxin-related Indole Acetic Acid (*IAA*) gene. This method was also used to verify the existence of the *M. marquandi* SGSF043 gene in the bud tissue.

### 4.10. Statistical Analysis

Microsoft Excel and SPSS 27 were used for data processing and statistical analysis, and the GraphPad Prism 10.0, Origin 2024 and PowerPoint software packages were used to form data graphs. The principal component analysis (PCA) diagram was drawn using the cloud tool platform of Shanghai Meiji Biological Co., LTD. The LSD test in one-way ANOVA was used to test the significance of differences among all groups at *p* < 0.05, * *p* < 0.05, ** *p* < 0.01, *** *p* < 0.001, **** *p* < 0.0001. In addition, Adobe Photoshop 2024 software was used for drawing.

## 5. Conclusions

The seed coating agent made from *M. marquandii* SGSF043 had significant positive effects on Chinese cabbage seeds, which not only enhanced the germination vitality of seeds but also promoted the growth of the lateral roots of young buds. In addition, the seed coating agent can also enhance the activity of antioxidant enzymes and hormone levels in different parts of Chinese cabbage buds. At the molecular level, it can significantly increase the relative expression levels of the chlorophyll-synthesis-related *BrPAO* gene; Abscisic Acid-related *BnNCED3* gene; *BrSAG12* gene; *IAA* gene; antioxidant-enzyme-related *BrSOD*, *BrPOD* and *BrCAT* genes; and Jasmonic acid (JA)-related *MYC3* gene in leaves. At the same time, the relative expression levels of the *ARF7* and *ARF19* genes promoting lateral root growth in the root were significantly increased, and the expression levels of antioxidant oxidase genes and ABA- and Jasmonic acid (JA)-related genes in the roots were generally downregulated, which maintained the antioxidant system and hormone balance of Chinese cabbage buds. These results indicated that *M. marquandii* SGSF043 coating agents promoted the early growth of Chinese cabbage and enhanced its immune capacity, partially activating the growth-promoting defense system of Chinese cabbage. This study provides a new perspective for future application under biotic and abiotic stress conditions and provides a new strategy for the development of ecological agriculture and biotechnology.

## Figures and Tables

**Figure 1 plants-14-00058-f001:**
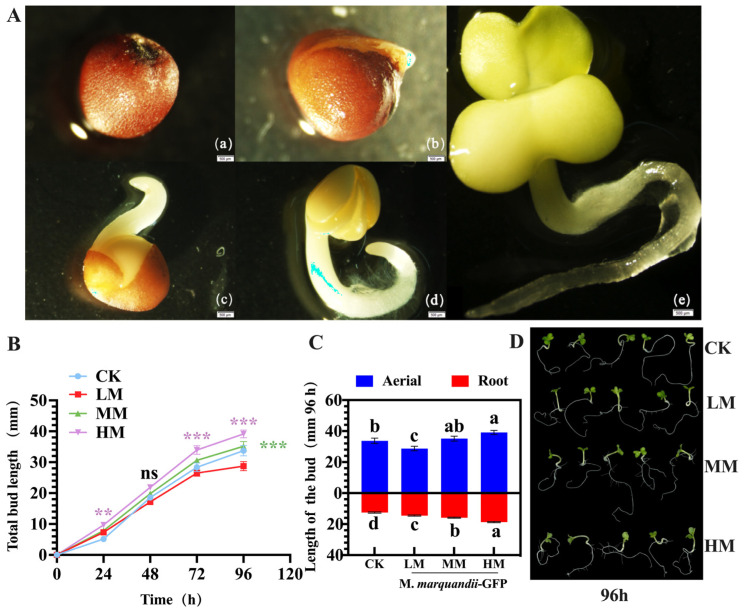
The whole process of cabbage seed germination. (**A**) and the effect of *M. marquandii* SGSF043 on germination of cabbage (**B**–**D**). Note: (**A**) (**a**) The imbibition stage, (**b**) the germination stage, (**c**,**d**) the germination stage, (**e**) the budding stage. (**B**–**D**) LM: the coating treatment of *M. marquandii* SGSF043 with 1 × 10^6^ CFU/mL; MM: the coating treatment of *M. marquandii* SGSF043 with 1 × 10^7^ CFU/mL; HM: the coating treatment of *M. marquandii* SGSF043 with 1 × 10^8^ CFU/mL. Control: PDB liquid medium was coated with blank control. * indicates a difference between different inoculation treatments (** *p* < 0.01; *** *p* < 0.001). a–d represent significant differences between different treatment groups (*p* < 0.05).

**Figure 2 plants-14-00058-f002:**
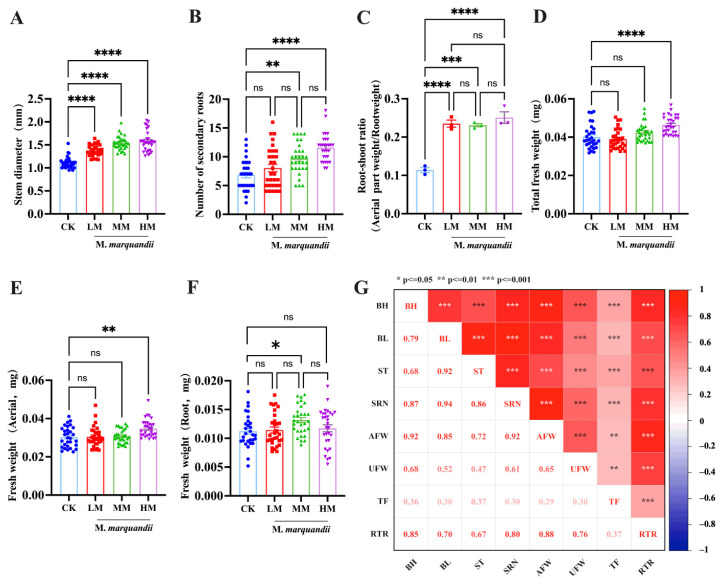
Effect of *M. marquandii* SGSF043 on secondary roots and the seedling biomass of the cabbage after 96 h. (**A**) Stem diameter of buds. (**B**) Number of secondary roots. (**C**) Root-shoot ratio (Aeria part weight/Root weight). (**D**) Total fresh weight of buds. (**E**) Fresh weight of aerial parts of buds. (**F**) Fresh weight of roots of buds. The result is the mean ± standard deviation (*n* = 30). * indicates a difference between different inoculation treatments (* *p* < 0.05; ** *p* < 0.01; *** *p* < 0.001; **** *p* < 0.0001). (**G**) Correlation heat map of each index. BH is the bud height, BL is bud root length, ST is stem diameter, SRN is number of secondary roots, AFW is above-ground fresh weight, UFW is below-ground fresh weight, TF is total fresh weight and RTR is root–shoot ratio (below-ground dry weight/above-ground dry weight). The result is the mean ± standard deviation (*n* = 30). * indicates a difference between different inoculation treatments (* *p* ≤ 0.05; ** *p* ≤ 0.01; *** *p* ≤ 0.001).

**Figure 3 plants-14-00058-f003:**
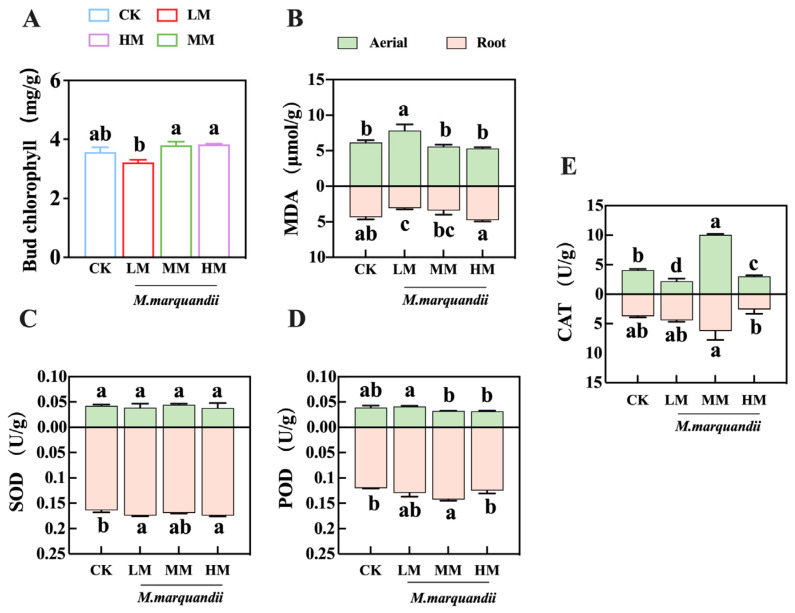
Effect of *M. marquandii* SGSF043 on chlorophyll, MDA contents and antioxidant enzyme activities of the cabbage buds after 96 h. The result is the mean ± standard deviation of three repeats (*n* = 3). (**A**) ChlorophyII content of leaves in different treatment groups. (**B**) MDA content in the aerial parts and roots of buds. (**C**) SOD activity in the aerial parts and roots of buds. (**D**) POD activity in the aerial parts and roots of buds. (**E**) CAT activity in the aerial parts and roots of buds. a–d represent significant differences between different treatment groups (*p* < 0.05).

**Figure 4 plants-14-00058-f004:**
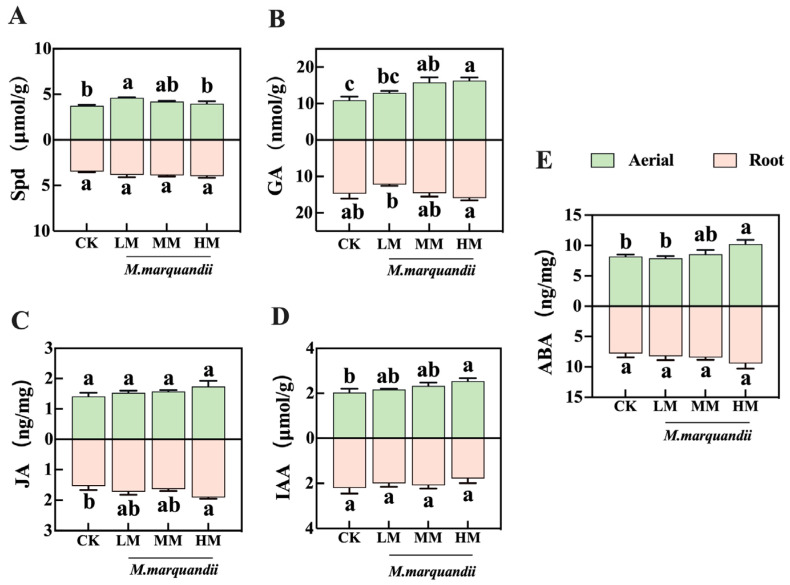
Effect of *M. marquandii* SGSF043 on the seedling phytohormone of the cabbage after 96 h. The result is the mean ± standard deviation (*n* = 3). (**A**) Spermidine content in the aerial parts and roots of buds. (**B**) GA content in the aerial parts and roots of buds. (**C**) JA content in the aerial parts and roots of buds. (**D**) IAA content in the aerial parts and roots of buds. (**E**) ABA content in the aerial parts and roots of buds. a–c represent significant differences between different treatment groups (*p* < 0.05).

**Figure 5 plants-14-00058-f005:**
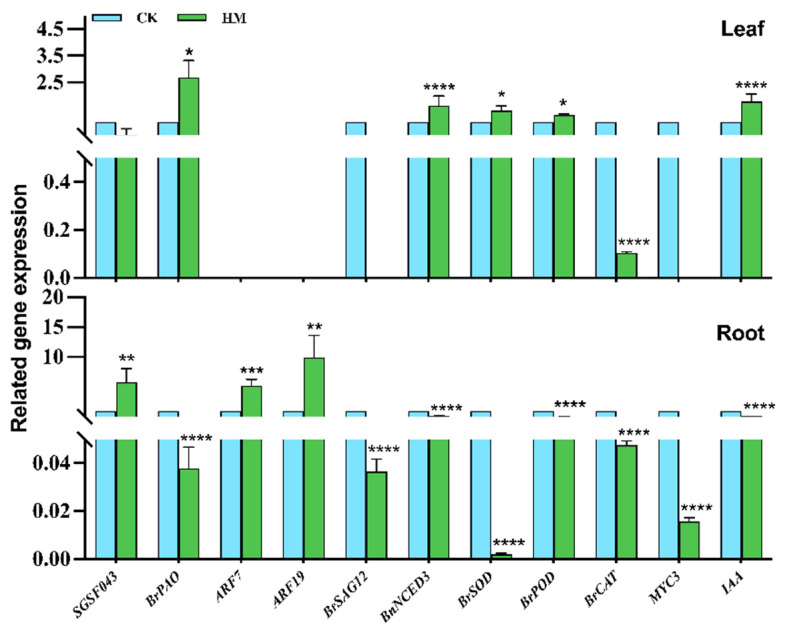
Expression of functional genes related to Chinese cabbage after 96 h. The result is the mean ± standard deviation of three repeats (*n* = 3). (CK) Expression of genes in control group without any treatment. (HM) Expression of genes in high concentration treatment group with *M. marquandii* SGSF043 at a concentration of 10^8^ CFU/mL. * indicates a difference between different inoculation treatments when the same gene is treated (* *p* < 0.05; ** *p* < 0.01; *** *p* < 0.001; **** *p* < 0.0001).

**Figure 6 plants-14-00058-f006:**
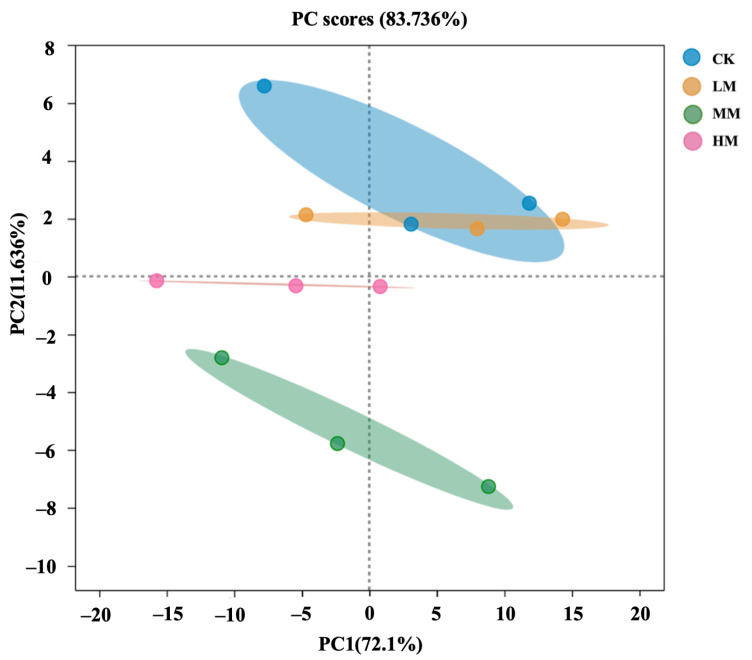
Principal component analysis of cabbage sprouts after 96 h.

**Figure 7 plants-14-00058-f007:**
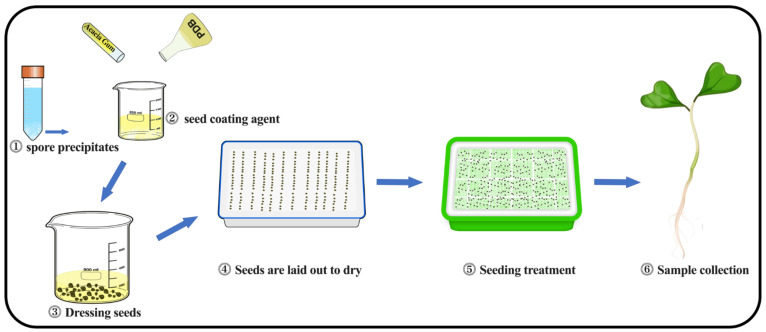
Preparation and budding process of coating agent *M. marquandii* SGSF043.

**Table 1 plants-14-00058-t001:** Effects of M. marquandii SGSF043 on seed germination of Chinese cabbage.

Treatment	Fungal Spore Content(/mL)	Germination Time/h	Germination Potential/%	Germination Rate/%	Germination Index	Bud Activity Index
*M. marquandii* SGSF043	LM (1 × 10^6^)	48	100.00 ± 0.00 ^a^	100.00 ± 0.00 ^a^	15.00 ± 0.00 ^a^	311.46 ± 0.1 ^c^
	MM (1 × 10^7^)	48	100.00 ± 0.00 ^a^	100.00 ± 0.00 ^a^	15.00 ± 0.00 ^a^	343.88 ± 0.03 ^b^
	HM (1 × 10^8^)	48	100.00 ± 0.00 ^a^	100.00 ± 0.00 ^a^	15.00 ± 0.00 ^a^	393.22 ± 0.02 ^a^
CK (PDB)	——	48	98.75 ± 0.04 ^a^	98.75 ± 0.04 ^a^	14.81 ± 0.53 ^a^	337.78 ± 0.04 ^bc^

The results are presented as the means ± standard deviations of three replicates. a–c represent significant differences between different treatment groups (*p* < 0.05).

## Data Availability

The data presented in this study are available on reasonable request from the corresponding author.

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
