# Peer review of "Effects of Marquandomyces marquandii SGSF043 on the Germination Activity of Chinese Cabbage Seeds: Evidence from Phenotypic Indicators, Stress Resistance Indicators, Hormones and Functional Genes"

_plants, 2024, doi:10.3390/plants14010058_

Round 1
Reviewer 1 Report
Comments and Suggestions for Authors
In this study, the authors investigated the influence of the fungus Marquandomyces marquandii SGSF043 on the germination of Chinese cabbage seeds and seedlings. The study was conducted using very complex phenotypic, biochemical, and molecular methods to get a deeper picture of the interaction between seedlings and the fungus. The experiment was well-designed and interesting results were obtained. However, the manuscript is poorly written in some parts.
The first sentence in the abstract is unnecessary.
Lines 44-45 Inhibit the growth and development of pathogenic and improve plant disease resistance-I guess this is a mistake
Line 57-66-rewrite the text
Line 109-110 However, there are few reports on the mechanism of this fungus in promoting plant growth and improving plant immunity. -add references
Line 110-112 How do you think you selected the fungal strain from the leaves of different plants?
Line 121 against biological and abiotic stresses -this part of the experiment was not performed in the study and therefore cannot be part of your description of the objectives
Line 126-add manufacturer
Line 127-Add how the test strain was isolated and identified
Line 133, 145-viable bacteria number -spore
Lines 135-143, 150-156, 189-195, 205-211, 215-216 -Rewrite the text according to the way methods are written in scientific papers
Line 137-Did the gum Arabic have a final concentration of 10% after mixing with the spore suspension?
Line 185 (ground dry weight/ground dry weight)?
Line 189-192-emphasize what this method was used for and how the total chlorophyll content was calculated and expressed
Lines 193-199-Add at the end of this part of the text how the MDA was calculated and expressed
Lines 200-203-Briefly describe all methods used to determine the enzyme activity and write how the enzyme activities were calculated and expressed
Table 1-The germination parameters in Table 1 do not match those described in the methods
Line 277, 318 when the same gene is treated-delete
Line 284-differences -add in secondary roots
Change the title of Figures 3 and 4 to match the data shown
Line 326-327 with the increase of the concentration of seed coating agent-add fungal
Figure 4-Enzyme activity, not content
Line 356, -BM?
Lines 392-393 ARF7 and ARF9 genes related to lateral root germination in Chinese cabbage can regulate lateral root germination in Chinese cab-rewrite
Lines 501-504 The effect of low concentration LM on the content of spermidine was the greatest, the effect of medium concentration MM and high concentration HM on the content of GA was the greatest, and the effect of high concentration HM on leaf IAA and ABA was the greatest—rewrite the sentence
-
Author Response
Comments 1: The first sentence in the abstract is unnecessary.
Response 1:Thank you for pointing this out. We agree with this comment. Therefore, We have deleted the first sentence.In lines 15-16.
Comments 2: Lines 44-45 Inhibit the growth and development of pathogenic and improve plant disease resistance-I guess this is a mistake
Response 2: Thank you for pointing this out. Therefore, We have changed :“For example, Flavobacteria and Sphingosphinomonas maintain strong metabolic capacity and are in a favorable position in competition with pathogens. It can inhibit the infection of pathogenic”.In line 49.
Comments 3:Line 57-66-rewrite the text
Response 3: Thank you for your guidance. Because we do not know whether the content needs to be rewritten or the language logic needs to be rewritten. We changed the logic. Hope for further guidance.In lines60-67.
Comments 4:Line 109-110 However, there are few reports on the mechanism of this fungus in promoting plant growth and improving plant immunity. -add references
Response 4: Thank you for your guidance. Marquandomyces marquandii used in this experiment is mainly used in the extraction of secondary metabolites and the repair of pesticide metabolites, etc. However, among the literature materials on promoting plant growth and improving plant immunity, we only found one article on the promotion of plant growth by this fungus. Therefore, our work wanted to carry out further research on the Marquandomyces marquandii in the future, so we organized this sentence.
Comments 5:Line 110-112 How do you think you selected the fungal strain from the leaves of different plants?
Response 5: Thanks to your guidance, we have supplemented the main sources of Marquandomyces marquandii in our materials and methods. It is mainly isolated from forest litter.
Comments 6:Line 121 against biological and abiotic stresses -this part of the experiment was not performed in the study and therefore cannot be part of your description of the objectives
Response 6: Thank you very much for your guidance. We think you are very strict and we have removed the sentence you pointed out.In lines128-129.
Comments 7:Line 126-add manufacturer
Response 7:Thank you for pointing this out.Because the position of the manuscript is a little different from the position you pointed out. Are you talking about the supplier of Chinese cabbage for the test? We added the name of the brand of Chinese cabbage for the test: Shandong Shouhe, China. Hope to get your guidance.In lines134-135.
Comments 8:Line 127-Add how the test strain was isolated and identified
Response 8:For this test fungal strain we have added this explanation:
- marquandiiSGSF043 is isolated from the litter in the Nanwenghe National Nature Reserve in Daxing'anling region, Heilongjiang Province. The main plant species are Tilia amurensisRupr., Quercus mongolica Fisch. Ex Turcz, Fraxinus mandshurica Rupr., etc. The litter is divided into undecomposed layer, semi-decomposed layer and decomposed layer from top to bottom. After collecting the samples in layers, we brought them back to the laboratory and place them in a cool place for natural air drying for separation and purification. The above strains are preserved in the strain collection center of Wuhan University, CCTCC No. M2020555.In lines137-145.
- Comments 9:Line 133, 145-viable bacteria number -spore
- Response 9:Thank you for your guidance. We have corrected the error. Should be the number of viable fungi.In line141.
-
Comments 10:Lines 135-143, 150-156, 189-195, 205-211, 215-216 -Rewrite the text according to the way methods are written in scientific papers
-
Response 10:Thank you for your guidance. We have reorganized and corrected the method according to your suggestion.In line 145-151,153-163,212-219,252-260.
-
Comments 11:Line 137-Did the gum Arabic have a final concentration of 10% after mixing with the spore suspension?
-
Response 11:Thank you for pointing this out.The spore suspension obtained by centrifugation was repeatedly cleaned with sterile water, and finally the precipitated spore suspension was poured into PDB at a fixed volume to ensure that the concentration of acacia in the treatment group and the control group was 10%. It was found that when the concentration of gelatin was 10%, the distribution of spores in seed coating was very uniform. This concentration is therefore chosen for coating.
We will "The seed coating agent was 10% coating agent prepared by adding Arabic gum to spore suspensions of different concentrations. ”Modified to: "Add Arabic gum to different spore suspensions, and finally use PDB to set the seed coating agent to 10% coating agent", do you think this is rigorous?
-
Comments 12:Line 185 (ground dry weight/ground dry weight)?
-
Response 12:Thank you for pointing it out. Sorry for our mistake. The correct spelling should be: "underground dry weight/above ground dry weight".In lines197-198.
-
Comments 13:Line 189-192-emphasize what this method was used for and how the total chlorophyll content was calculated and expressed
-
Response 13:Thank you very much for your guidance, we have added the content. See 2.6 for details.In line 199.
-
Comments 14:Lines 193-199-Add at the end of this part of the text how the MDA was calculated and expressed
-
Response 14:Thank you very much for your guidance, we have added the content. See 2.6 for details.In line 199.
-
Comments 15:Lines 200-203-Briefly describe all methods used to determine the enzyme activity and write how the enzyme activities were calculated and expressed
Table 1-The germination parameters in Table 1 do not match those described in the methods
-
Response 15:Thank you for your guidance. We have added measurement methods and calculation formulas to the manuscript.In line202,2.5 for details.
We have unified the corresponding indicators in Table 1.line 293.
-
Comments 16:Line 277, 318 when the same gene is treated-delete
-
Response 16:Thank you for your guidance. We have deleted the useless content.line307,349.
-
Comments 17:Line 284-differences -add in secondary roots
Change the title of Figures 3 and 4 to match the data shown
-
Response 17:Thank you very much for your guidance. We changed the headings for Figures 3 and 4. As follows:
Figures 3:Effect of M. marquandii SGSF043 on secondary roots and the seedling biomass of the cabbage in 96h.In line340.
Figures 4:Effect of M. marquandii SGSF043 on contents of Chlorophyll,MDA and Antioxidant Enzymes of the cabbage buds in 96h.In line381.
-
Comments 18:Line 326-327 with the increase of the concentration of seed coating agent-add fungal
Figure 4-Enzyme activity, not content
-
Response 18:Thank you for pointing it out. We changed the meaning to increase as the concentration of the fungus increases.In lines346-347.
And we changed the content in the graph to activity.In line 369.
-
Comments 19:Line 356, -BM?
-
Response 19:Thank you for your guidance. We changed the "BM" in the article to "HM" for processing.
-
Comments 20:Lines 392-393 ARF7 and ARF9 genes related to lateral root germination in Chinese cabbage can regulate lateral root germination in Chinese cab-rewrite
-
Response 20:We are very sorry that ARF19 was written as ARF9. I am really sorry that it is not rigorous enough.In lines 421-424.
-
Comments 21:Lines 501-504 The effect of low concentration LM on the content of spermidine was the greatest, the effect of medium concentration MM and high concentration HM on the content of GA was the greatest, and the effect of high concentration HM on leaf IAA and ABA was the greatest—rewrite the sentence
-
Response 21: Thank you for your guidance. Another review expert asked the same question as you. We revised this sentence to: LM treatment had the greatest effect on the content of spermidine in leaves, QM and BM treatment had the greatest effect on the content of Gibberellin in leaves, and BM treatment had the greatest effect on the content of Indole Acetic Acid and abolic acid in leaves.In lines544-547.

Reviewer 2 Report
Comments and Suggestions for Authors
Manuscript ID: plants-3261234
Article Title: Effects of Marquandomyces Marquandii SGSF043 on Improving the Germination Activity of Chinese Cabbage Seeds: Evidence from Phenotypic Indicators, Stress Resistance Indicators, Hormones and Functional Genes
The manuscript describes the effect of coating the seeds of Chinese Cabbage with the fungal strain Marquandomyces marquandii SGSF043. The presence of the fungus improves germination and other parameters such as antioxidant response, phytohormone production, growth, and stress tolerance in Chinese Cabbage.
Overall, the research described in the manuscript is interesting, the selected methodological approaches are adequate, and the results are consistent. However, the manuscript has several formatting problems and needs to be reviewed regarding different aspects related to the redaction in English. Authors need to address all the following commentaries.
Commentaries:
In line 2, correct scientific name of the fungal strain in title
In lines 14-15, it is not clear which microorganism in employed in the study “Metarhizium anisopliae” or “Marquandomyces marquandii” please review
Line 20 and 21, Check “antioxidase system” it is correct, could be “antioxidant system”
Line 23, define the acronyms of the hormones “IAA, GA and ABA”
Line 29, it is not clear which microorganism in employed in the study “Metarhizium anisopliae” or “Marquandomyces marquandii” please review
Lines 33 and 34, eliminate unnecessary period in “Brassica rapa.”
Line 36 and 39, eliminate extra space after references
Line 50, define the acronym “GA” first tome used in the manuscript
Line 51, check redaction in “and its phosphate to dissolve may help promote growth of cabbage” it is difficult to understand
Lines 52, 56 and 57, eliminate extra space after references
Line 58, use lowercase for “Based”
Line 63, add missing space in “carcasses[14].Based”
Line 66, eliminate extra space in “years[13,15]. Treatment”
Line 68, correct scientific name in “Arabidopsis Thaliana”
Lines 71-73, check redaction in “Vigorous growth of plants can provide a more stable habitat for Metarhizium spp., while the parasite of Metarhizium spp. can further protect the healthy growth of plants” it is confusing
Line 77, complement with the pathogens that cause each plant disease
Line 81, describe better or add some examples of “certain effects on plant hormones"
Line 84, correct scientific name in “Metarhizium Roberts”
Line 85, correct scientific name in “Arabidopsis Thaliana”
Line 86, define the acronym “IAA” first tome used in the manuscript
Line 91, give examples of “immune-related genes”
Line 96, define the acronym “JA and SA” first tome used in the manuscript
Line 118-121, describe in better detail of give examples of which biological or abiotic stressors will be considered
Lines 132-133, the study evaluates a fungal strain, not bacteria
Lines 138 and 139 “gum Arabic” could be “Arabic gum”
Line 143, which means number in circles?
Lines 145-147, define which means “LM, MM, HM, and CK” to avoid confusing in subsequent descriptions
Line 145, the study evaluates a fungal strain, not bacteria
Line 152, add missing space in “50mL”
Line 187, define the acronym “MDA” first tome used in the manuscript
Line 191-192, add missing spaces “0.45µmol” and in “665nm, 649nm and 470nm”
Line 192, describe better why measure the absorbance
Lines 193-199, add missing spaces in whole paragraph
Line 199, describe better why measure the absorbance
Lines 205-211, restructure this paragraph, it is very repetitive and has formatting errors
Line 217, add missing period in “United States) The reverse”
Line 222, complement with the country “Jilin”
Lines 277-278, add a space to separate the paragraph from figure caption
In Figure 2, If the process was recorded, indicate at what time each process occurs, and compare with the control
Line 281, add missing space in “1:TableS2).Stem”
Line 288, “24.92%, 32.16% and 53.68%” could be “24.92, 32.16 and 53.68%”
Line 291, in HM means high-concentration inoculum “just us HM”
Line 299, “118.18%, 109.09% and 127.27%” could be “118.18, 109.09 and 127.27%”
Line 325, add missing spaces and periods in “groups(P<0.05),The”
Line 327, in LM means low-concentration inoculum “just us LM”
Lines 328, 332, 335, 338, 340, 342, add missing spaces in all paragraph
Line 337, in MM means medium-concentration inoculum “just us MM”
Line 343, catalase was previously reported as “CAT”
Line 424, add missing space in “96h.the”
Author Response
Comments 1: In line 2, correct scientific name of the fungal strain in title
Response 1: Thank you for pointing this out. We agree with this comment. Therefore, we have changed the title to“Effects of Marquandomyces marquandii SGSF043 on Improving the Germination Activity of Chinese Cabbage Seeds:Evidence from Phenotypic Indicators, Stress Resistance Indicators, Hormones and Functional Genes”.See line 2 for details.
Comments 2: In lines 14-15, it is not clear which microorganism in employed in the study “Metarhizium anisopliae” or “Marquandomyces marquandii” please review
Response 2: Thank you for pointing this out. We agree with this comment. In order to unify the statement, we changed “Metarhizium anisopliae” to “Marquandomyces marquandii ”.In lines15-16.
Comments 3: Line 20 and 21, Check “antioxidase system” it is correct, could be “antioxidant system”
Response 3:Thank you for pointing this out. We agree with this comment. We have, accordingly, changed.to emphasize this point. In lines 22-23.
Comments 4: Line 23, define the acronyms of the hormones “IAA, GA and ABA”
Response 4:Thank you for pointing this out. We agree with this comment. We have included the full names of each hormone in the article. I don't know if this change meets the requirements, and I hope to get your guidance again.“the effect of M. marquandii SGSF043 on the leaf hormone Indole Acetic Acid(IAA), Gibberellic Acid(GA) and Abscisic Acid(ABA) of Chinese cabbage seedlings was significantly higher than that of other treatment groups, indicating that the strain could optimize the level of plant hormones”.Inlines 25.
Comments 5: Line 29, it is not clear which microorganism in employed in the study “Metarhizium anisopliae” or “Marquandomyces marquandii” please review
Response 5:Thank you for pointing this out. We agree with this comment. In order to unify the statement, we changed “Metarhizium anisopliae” to “Marquandomyces marquandii ”.In line32.
Comments 6: Lines 33 and 34, eliminate unnecessary period in “Brassica rapa.”
Response 6:Thank you for pointing this out. We agree with this comment. Therefore, we have changed.In line 38.
Comments 7: Line 36 and 39, eliminate extra space after references
Response 7:Thank you for pointing this out. We agree with this comment. Therefore, we have changed.In line36-line39.
Comments 8: Line 50, define the acronym “GA” first tome used in the manuscript
Response 8:Thank you for pointing this out. We agree with this comment. The references cited in the article refer to GA4, GA8, GA9, GA19, and GA20. Therefore, we use “Gibberellin ” to replace the definition of “GA”. Do you think this modification is correct? In line 55.
Comments 9: Line 51, check redaction in “and its phosphate to dissolve may help promote growth of cabbage” it is difficult to understand
Response 9:Thank you for pointing this out. We agree with this comment. Therefore, we have changed it.“The ability of Gibberellin secreted by Bacillus amylolyticus and its phosphate to dissolve may help promote growth of cabbage, radish, tomato and mustard greens,and improve the salt tolerance of plants[9] ”
Comments 10: Lines 52, 56 and 57, eliminate extra space after references
Response 10:Thank you for pointing this out. We agree with this comment. Therefore, we have changed it.In line 62.
Comments 11: Line 58, use lowercase for “Based”
Response 11:Thank you for pointing this out. We agree with this comment. Therefore, We've changed "Based" to "based."In line 68.
Comments 12: Line 63, add missing space in “carcasses[14].Based”
Response 12:Thank you for pointing this out. I/We agree with this comment. Therefore, We've changed "Based" to ", based."In line 68.
Comments 13: Line 66, eliminate extra space in “years[13,15]. Treatment
Response 13:Thank you for pointing this out. We agree with this comment. Therefore, we have been deleted extra space.
Comments 14: Line 68, correct scientific name in “Arabidopsis Thaliana”
Response 14:Thank you for pointing this out. We agree with this comment. Therefore, We've changed "Arabidopsis Thaliana", Arabidopsis thaliana"In line 74.
Comments 15: Lines 71-73, check redaction in “Vigorous growth of plants can provide a more stable habitat for Metarhizium spp., while the parasite of Metarhizium spp. can further protect the healthy growth of plants” it is confusing
Response 15:Thank you for pointing this out. We agree with this comment. Therefore, we have changed it.After consulting the cited literature, we changed this sentence to: “Plants can provide a more stable habitat for Metarhizium spp., while the parasite of Metarhizium spp. can further protect the healthy growth of plants, such as microorganisms can protect plants by dissolving inorganic nutrients or acting as biological control agents for plant pathogens.”In line77.
Comments 16: Line 77, complement with the pathogens that cause each plant disease
Response 16:Thank you for pointing this out. We agree with this comment. Therefore, We finished the replenishment.As follows:In lines 82-87.
For example, Metarhizium spp. has inhibitory effects on Fusarium Oxysporum Schl, Physoderma maydis Miyabe, Fusarium graminearum, Fusarium oxysporum f.sp. Lycopersici Snyder et Hansen and Botrytis cinerea Pers. At present, research on how Metarhizium spp. inhibits the growth of plant pathogens is still in its infancy. And the molecular mechanisms of the interaction between the host plant-Metarhizium spp. symbiosis and the pathogen also need further study.
Comments 17: Line 81, describe better or add some examples of “certain effects on plant hormones"
Response 17:Thank you for pointing this out. We agree with this comment. Therefore, We supplemented the index parameters of Metarhizium anisopliae regulating plant hormone levels to promote plant growth.In lines87-91.
Comments 18: Line 84, correct scientific name in “Metarhizium Roberts”
Response 18:Thank you for pointing this out. We agree with this comment. Therefore, We've changed "Metarhizium Roberts" to "Metarhizium robertsii."In line 90.
Comments 19: Line 85, correct scientific name in “Arabidopsis Thaliana”
Response 19:Thank you for pointing this out. We agree with this comment. Therefore, We've changed "Arabidopsis Thaliana", Arabidopsis thaliana"In line 92.
Comments 20: Line 86, define the acronym “IAA” first tome used in the manuscript
Response 20:Thank you for pointing this out. We agree with this comment. Therefore, we have added the full name of the IAA explanation.
Comments 21: Line 91, give examples of “immune-related genes”
Response 21:Thank you for pointing this out. We agree with this comment. Therefore, We have added examples of relevant immune genes.In line99.
Comments 22: Line 96, define the acronym “JA and SA” first tome used in the manuscript
Response 22:Thank you for pointing this out. We agree with this comment. Therefore, we have added the full name of the “JA and SA” explanation.
Comments 23: Line 118-121, describe in better detail of give examples of which biological or abiotic stressors will be considered
Response 23:Thank you for your question. In summary, biological and abiotic stress experiments were not involved in this study, combined with the suggestion of another reviewer. We have revised this sentence. "and enhance the "Immunity" of plants against biological and abiotic stresses. " is deleted. Hope to get your further guidance and suggestions.In line 129.
Comments 24: Lines 132-133, the study evaluates a fungal strain, not bacteria
Response 24:Thank you for pointing this out. We agree with this comment. Therefore, we have changed “bacteria”to “fungal spore”. In line 141.
Comments 25: Lines 138 and 139 “gum Arabic” could be “Arabic gum”
Response 25:Thank you for pointing this out. We agree with this comment. Therefore, we have changed “gum Arabic”to “Arabic gum”. In line 146.
Comments 26: Line 143, which means number in circles?
Response 26:Thank you for pointing this out. We agree with this comment. We matched the words to the corresponding circles.In lines 143-152.
Comments 27: Lines 145-147, define which means “LM, MM, HM, and CK” to avoid confusing in subsequent descriptions
Response 27:Thank you for pointing this out. We agree with this comment. Please give guidance and suggestions on the following changed content.“A total of 4 coating agents were set up in this experiment: The effective viable fungal count was 1×106 CFU/mL coating agent low concentration (LM), 1×107 CFU/mL coating agent medium concentration (MM), 1×108 CFU/mL coating agent high concentration (HM) and blank control (CK) treated with PDB medium directly.”
Comments 28: Line 145, the study evaluates a fungal strain, not bacteria
Response 28:Thank you for pointing this out. We agree with this comment. Therefore, we have changed “bacteria”to“fungal”.In line 154.
Comments 29: Line 152, add missing space in “50mL”
Response 29:Line 152, add missing space in “50mL”
Comments 30: Line 187, define the acronym “MDA” first tome used in the manuscript
Response 30:Thank you for pointing this out. We agree with this comment. Therefore, We have marked the meaning of MDA in the method.In line 205.
Comments 31: Line 191-192, add missing spaces “0.45µmol” and in “665nm, 649nm and 470nm”
Response 31:Thank you for pointing this out. We agree with this comment. Therefore, we have changed it.In line202-203.
Comments 32: Line 192, describe better why measure the absorbance
Response 32:Thank you for pointing this out. We agree with this comment. Therefore,We have added the significance of measuring absorbance in order to determine total chlorophyll content.In line 205.
Comments 33: Lines 193-199, add missing spaces in whole paragraph
Response 33:Thank you for pointing this out. We agree with this comment. Therefore,We have corrected the mistake.In lines201-205.
Comments 34: Line 199, describe better why measure the absorbance
Response 34:Thank you for pointing this out. We agree with this comment. The determination of absorbance is mainly to measure the MDA concentration through the formula, and then calculate the MDA content in plants.We have described it in the manuscript.In lines 213-214.
Comments 35: Lines 205-211, restructure this paragraph, it is very repetitive and has formatting errors
Response 35:Thank you for pointing this out. We agree with this comment. Therefore, we have changed it.As follows:
Following the manufacturer's guidelines, employ the following products from Beijing Boaotoda Technology Co., LTD., all based in Beijing, China, to determine the levels of specific hormones in 96-hour-old Chinese cabbage buds (leaves, roots):
Phytospermine (Product Code: TOPEL30097)
Gibberellic Acid (GA) (Product Code: TOPEL03457)
Jasmonic Acid (JA) (Product Code: TOPEL30282)
Indole Acetic Acid (IAA) (Product Code: TOPEL30282)
Abscisic Acid (ABA) (Product Code: TOPEL03473).In lines 228-235.
Response 35:Thank you for pointing this out. We agree with this comment. Therefore, we have changed it.As follows:
Following the manufacturer's guidelines, employ the following products from Beijing Boaotoda Technology Co., LTD., all based in Beijing, China, to determine the levels of specific hormones in 96-hour-old Chinese cabbage buds (leaves, roots):
Phytospermine (Product Code: TOPEL30097);Gibberellic Acid (GA) (Product Code: TOPEL03457);Jasmonic Acid (JA) (Product Code: TOPEL30282);Indole Acetic Acid (IAA) (Product Code: TOPEL30282);Abscisic Acid (ABA) (Product Code: TOPEL03473).In lines 228-235.
Comments 36: Line 217, add missing period in “United States) The reverse”
Response 36:Thank you for pointing this out. We agree with this comment. Therefore, we have changed it.In line231.
Comments 37: Line 222, complement with the country “Jilin”
Response 37:Thank you for pointing this out. We agree with this comment. Therefore, we have changed it.Put "Jilin Province, China" in parentheses.In line 236.
Comments 38: Lines 277-278, add a space to separate the paragraph from figure caption
Response 38:Thank you for pointing this out. We agree with this comment. Therefore, we have changed it.In line298.
Comments 39: In Figure 2, If the process was recorded, indicate at what time each process occurs, and compare with the control
Response 39:Thank you for pointing this out. We agree with this comment. We have annotated the germination process and described it in conjunction with the description.In lines312,322-323.
Comments 40: Line 281, add missing space in “1:TableS2).Stem”
Response 40:Thank you for pointing this out. We agree with this comment. Therefore, we have changed it.In line 301.
Comments 41: Line 288, “24.92%, 32.16% and 53.68%” could be “24.92, 32.16 and 53.68%”
Response 41:Thank you for pointing this out. We agree with this comment. Therefore, we have changed it.In lines 302 and 308.
Comments 42:Line 291, in HM means high-concentration inoculum “just us HM”
Response 42:Thank you for pointing this out. We agree with this comment. Therefore, We have corrected the mistake.As shown below:
In the peroxidase (POD) treatment groups (Figure 4D), the above-ground data showed that there were significant differences between LM treatment group and the control group and MM and HM treatment groups (P<0.05). There was significant difference between LM treatment group and HM treatment group and control group(P<0.05). In lines351-369.
Comments 43: Line 299, “118.18%, 109.09% and 127.27%” could be “118.18, 109.09 and 127.27%”
Response 43:Thank you for pointing this out. We agree with this comment. Therefore, we have changed it.In lines 318 and 319.
Comments 44:Line 325, add missing spaces and periods in “groups(P<0.05),The”
Response 44:Thank you for pointing this out. We agree with this comment. Therefore, we have changed it.In line 346.
Comments 45:Line 327, in LM means low-concentration inoculum “just us LM”
Response 45:Thank you for pointing this out. We agree with this comment. Therefore, We have corrected the mistake.As shown below:
In the peroxidase (POD) treatment groups (Figure 4D), the above-ground data showed that there were significant differences between LM treatment group and the control group and MM and HM treatment groups (P<0.05). There was significant difference between LM treatment group and HM treatment group and control group(P<0.05). In lines351-369.
Comments 46:Lines 328, 332, 335, 338, 340, 342, add missing spaces in all paragraph
Response 46:Thank you for pointing this out. We agree with this comment. Therefore, we have changed it.In lines 351-368.
Comments 47:Line 337, in MM means medium-concentration inoculum “just us MM”
Response 47:Thank you for pointing this out. We agree with this comment. Therefore, We have corrected the mistake.As shown below:
In the peroxidase (POD) treatment groups (Figure 4D), the above-ground data showed that there were significant differences between LM treatment group and the control group and MM and HM treatment groups (P<0.05). There was significant difference between LM treatment group and HM treatment group and control group(P<0.05). In lines351-369.
Comments 48:Line 343, catalase was previously reported as “CAT”
Response 48:Thank you for pointing this out. We agree with this comment. Therefore, we have deleted“(CAT)”.In line 373.
Comments 49:Line 424, add missing space in “96h.the”
Response 49:Thank you for pointing this out. We agree with this comment. Therefore, we have changed it.In line 453.

Round 2
Reviewer 1 Report
Comments and Suggestions for Authors
Comments 2: Lines 44-45 Inhibit the growth and development of pathogenic and improve plant disease resistance-I guess this is a mistake
Response 2: Thank you for pointing this out. Therefore, We have changed :“For example, Flavobacteria and Sphingosphinomonas maintain strong metabolic capacity and are in a favorable position in competition with pathogens. It can inhibit the infection of pathogenic”.In line 49.
Response R: Pathogenic is an adjective. It should be pathogenic microorganisms
Comments 4:Line 109-110 However, there are few reports on the mechanism of this fungus in promoting plant growth and improving plant immunity. -add references
Response 4: Thank you for your guidance. Marquandomyces marquandii used in this experiment is mainly used in the extraction of secondary metabolites and the repair of pesticide metabolites, etc. However, among the literature materials on promoting plant growth and improving plant immunity, we only found one article on the promotion of plant growth by this fungus. Therefore, our work wanted to carry out further research on the Marquandomyces marquandii in the future, so we organized this sentence.
Response R: If you found only one reference, rewrite the sentence and add that one reference.
Comments 5:Line 110-112 How do you think you selected the fungal strain from the leaves of different plants?
Response 5: Thanks to your guidance, we have supplemented the main sources of Marquandomyces marquandii in our materials and methods. It is mainly isolated from forest litter.
Response R: You have not changed the text in the manuscript. Line 114
Comments 7:Line 126-add manufacturer
Response 7:Thank you for pointing this out.Because the position of the manuscript is a little different from the position you pointed out. Are you talking about the supplier of Chinese cabbage for the test? We added the name of the brand of Chinese cabbage for the test: Shandong Shouhe, China. Hope to get your guidance.In lines134-135.
Response R: I meant the seed producer. In the manuscript there is only the brand but not the producer. Line 129
Comments 8:Line 127-Add how the test strain was isolated and identified
Response 8:For this test fungal strain we have added this explanation:
marquandiiSGSF043 is isolated from the litter in the Nanwenghe National Nature Reserve in Daxing'anling region, Heilongjiang Province. The main plant species are Tilia amurensisRupr., Quercus mongolica Fisch. Ex Turcz, Fraxinus mandshurica Rupr., etc. The litter is divided into undecomposed layer, semi-decomposed layer and decomposed layer from top to bottom. After collecting the samples in layers, we brought them back to the laboratory and place them in a cool place for natural air drying for separation and purification. The above strains are preserved in the strain collection center of Wuhan University, CCTCC No. M2020555.In lines137-145.
Response R: Rewrite this passage using the past passive voice.
The main plant species are Tilia amurensisRupr., Quercus mongolica Fisch. Ex Turcz, Fraxinus mandshurica Rupr., etc. I guess in the litter?
It is not written how the strain was identified.
Comments 10:Lines 135-143, 150-156, 189-195, 205-211, 215-216 -Rewrite the text according to the way methods are written in scientific papers
Response 10:Thank you for your guidance. We have reorganized and corrected the method according to your suggestion.In line 145-151,153-163,212-219,252-260.
Response R: Some parts of the text are still not well written.
Lines 154-155, 166-168, 204-211, 219-226, 232-244, 251-263, 269-279-Note that methods are always written in the past passive.
Comments 12:Line 185 (ground dry weight/ground dry weight)?
Response 12:Thank you for pointing it out. Sorry for our mistake. The correct spelling should be: "underground dry weight/above ground dry weight".In lines197-198.
Response R: Unchanged in the revised manuscript
Comments 17 Change the title of Figures 3 and 4 to match the data shown
Response 17:Thank you very much for your guidance. We changed the headings for Figures 3 and 4. As follows:
Figures 3:Effect of M. marquandii SGSF043 on secondary roots and the seedling biomass of the cabbage in 96h.In line340.
Figures 4:Effect of M. marquandii SGSF043 on contents of Chlorophyll,MDA and Antioxidant Enzymes of the cabbage buds in 96h.In line381
Response R: Fig 4 contents of Chlorophyll,MDA and Antioxidant Enzymes- change in chlorophyll and MDA contents and antioxidant enzyme activities
Comments 21:Lines 501-504 The effect of low concentration LM on the content of spermidine was the greatest, the effect of medium concentration MM and high concentration HM on the content of GA was the greatest, and the effect of high concentration HM on leaf IAA and ABA was the greatest—rewrite the sentence
Response 21: Thank you for your guidance. Another review expert asked the same question as you. We revised this sentence to: LM treatment had the greatest effect on the content of spermidine in leaves, QM and BM treatment had the greatest effect on the content of Gibberellin in leaves, and BM treatment had the greatest effect on the content of Indole Acetic Acid and abolic acid in leaves.In lines544-547.
Response R: Try to write this without repeating the words.
Comments on the Quality of English LanguageSome parts of the text are poorly written, and comments have been sent to the authors.
Author Response
- Comments 2:Lines 44-45 Inhibit the growth and development of pathogenic and improve plant disease resistance-I guess this is a mistake
Response 2: Thank you for pointing this out. Therefore, We have changed :“For example, Flavobacteria and Sphingosphinomonas maintain strong metabolic capacity and are in a favorable position in competition with pathogens. It can inhibit the infection of pathogenic”.In line 49.
Response R: Pathogenic is an adjective. It should be pathogenic microorganisms
Response:Thanks for your guidance, we have changed “pathogenic” to “pathogenic microorganisms”.In line 46.
- Comments 4: Line 109-110 However, there are few reports on the mechanism of this fungus in promoting plant growth and improving plant immunity. -add references
Response 4: Thank you for your guidance. Marquandomyces marquandii used in this experiment is mainly used in the extraction of secondary metabolites and the repair of pesticide metabolites, etc. However, among the literature materials on promoting plant growth and improving plant immunity, we only found one article on the promotion of plant growth by this fungus. Therefore, our work wanted to carry out further research on the Marquandomyces marquandii in the future, so we organized this sentence.
Response R: If you found only one reference, rewrite the sentence and add that one reference.
Response:Thanks to your suggestion, we have added the corresponding reference to the manuscript and revised the quote.In lines113-114,[30].
- Comments 5: Line 110-112 How do you think you selected the fungal strain from the leaves of different plants?
Response 5: Thanks to your guidance, we have supplemented the main sources of Marquandomyces marquandii in our materials and methods. It is mainly isolated from forest litter.
Response R: You have not changed the text in the manuscript. Line 114.
Response: Thank you for your guidance. We may have forgotten to save the revised manuscript. The source of the tested fungi has been briefly added in the first part of the manuscript, and the specific source is explained in the following materials and methods. In lines 115-126.
- Comments 7: Line 126-add manufacturer
Response 7:Thank you for pointing this out.Because the position of the manuscript is a little different from the position you pointed out. Are you talking about the supplier of Chinese cabbage for the test? We added the name of the brand of Chinese cabbage for the test: Shandong Shouhe, China. Hope to get your guidance.In lines134-135.
Response R: I meant the seed producer. In the manuscript there is only the brand but not the producer. Line 129
Response:We included seed producers in the article.“Shandong Shouhe,Shouguang Xinranran Horticulture Limited Company.”In line130.
- Comments 8: Line 127-Add how the test strain was isolated and identified
Response 8: For this test fungal strain we have added this explanation:
marquandiiSGSF043 is isolated from the litter in the Nanwenghe National Nature Reserve in Daxing'anling region, Heilongjiang Province. The main plant species are Tilia amurensisRupr., Quercus mongolica Fisch. Ex Turcz, Fraxinus mandshurica Rupr., etc. The litter is divided into undecomposed layer, semi-decomposed layer and decomposed layer from top to bottom. After collecting the samples in layers, we brought them back to the laboratory and place them in a cool place for natural air drying for separation and purification. The above strains are preserved in the strain collection center of Wuhan University, CCTCC No. M2020555.In lines137-145.
Response R: Rewrite this passage using the past passive voice.
The main plant species are Tilia amurensisRupr., Quercus mongolica Fisch. Ex Turcz, Fraxinus mandshurica Rupr., etc. I guess in the litter?
It is not written how the strain was identified.
Response:Thank you for your guidance, we agree with your suggestion, the changes are detailed in: lines140-148.And the corresponding references are added.
- Comments 10: Lines 135-143, 150-156, 189-195, 205-211, 215-216 -Rewrite the text according to the way methods are written in scientific papers
Response 10: Thank you for your guidance. We have reorganized and corrected the method according to your suggestion.In line 145-151,153-163,212-219,252-260.
Response R: Some parts of the text are still not well written.
Lines 154-155, 166-168, 204-211, 219-226, 232-244, 251-263, 269-279-Note that methods are always written in the past passive.
Response :Thank you for your guidance. We have modified the relevant materials and methods and the specific determination methods.In lines:113-315.
- Comments 12: Line 185 (ground dry weight/ground dry weight)?
Response 12: Thank you for pointing it out. Sorry for our mistake. The correct spelling should be: "underground dry weight/above ground dry weight".In lines197-198.
Response R: Unchanged in the revised manuscript
Response:Thank you for your guidance. We have changed "ground dry weight/ground dry weigh" to "underground dry weight/above ground dry weight".But maybe the number of rows we open are412,421.
- Comments 17 : Change the title of Figures 3 and 4 to match the data shown
Response 17: Thank you very much for your guidance. We changed the headings for Figures 3 and 4. As follows:
Figures 3:Effect of M. marquandii SGSF043 on secondary roots and the seedling biomass of the cabbage in 96h.In line340.
Figures 4:Effect of M. marquandii SGSF043 on contents of Chlorophyll,MDA and Antioxidant Enzymes of the cabbage buds in 96h.In line381
Response R: Fig 4 contents of Chlorophyll,MDA and Antioxidant Enzymes- change in chlorophyll and MDA contents and antioxidant enzyme activities
Response:Thank you for your guidance. We corrected the errors in the manuscript.“Figure 4. Effect of M. marquandii SGSF043 on chlorophyll,MDA contents and antioxidant enzymes activities of the cabbage buds in 96h. ”In line463.
- Comments 21: Lines 501-504 The effect of low concentration LM on the content of spermidine was the greatest, the effect of medium concentration MM and high concentration HM on the content of GA was the greatest, and the effect of high concentration HM on leaf IAA and ABA was the greatest—rewrite the sentence
Response 21: Thank you for your guidance. Another review expert asked the same question as you. We revised this sentence to: LM treatment had the greatest effect on the content of spermidine in leaves, QM and BM treatment had the greatest effect on the content of Gibberellin in leaves, and BM treatment had the greatest effect on the content of Indole Acetic Acid and abolic acid in leaves.In lines544-547.
Response R: Try to write this without repeating the words.
Response: Thank you for your guidance again. Or please ask the reviewer to mark specific mistakes.Hope to get the reviewer's guidance.We changed the sentence. As follows:“Low concentration LM most affected spermidine levels, medium concentration MM and high concentration HM most affected GA levels, and high concentration HM most influenced leaf IAA and ABA levels.”
Dear reviewer
Thank you for your careful guidance. In the future manuscript writing, we will definitely make more efforts to write the manuscript rigorously. Thank you very much for your hard work. Your guidance has been very helpful in improving our writing skills. We look forward to receiving your guidance again sometime. Thank you very much! I wish you a smooth work, a happy life and all the best!
Best wishes from Zheng Xu
December 2024.
Reviewer 2 Report
Comments and Suggestions for Authors
The authors adequately addressed all formatting problems and reviewed the redaction in English. I do not have additional commentaries.
Author Response
Dear reviewer
Thanks for the expert's careful guidance. In the course of revising the paper, we learned a lot about the precautions of writing. In the future writing, we will try to make the manuscript as rigorous as possible, looking forward to the next cooperation!
Best wishes From ZhengXu.